

# Spatiotemporal variability of light attenuation and net ecosystem metabolism in a back-barrier estuary

Neil K. Ganju[1], Jeremy M. Testa[2], Steven E. Suttles[1], Alfredo L. Aretxabaleta[1]

[1]U.S. Geological Survey, Woods Hole Coastal and Marine Science Center, Woods Hole, MA

[2]Chesapeake Biological Laboratory, University of Maryland Center for Environmental Science, Solomons, MD
*Correspondence to:* Neil K. Ganju (nganju@usgs.gov)

**Abstract.** The light climate in back-barrier estuaries is a strong control on phytoplankton and submerged aquatic vegetation (SAV) growth, and ultimately net ecosystem metabolism. However, quantifying the spatiotemporal variability of light attenuation and net ecosystem metabolism over seasonal timescales is difficult due to sampling

limitations and dynamic physical and biogeochemical processes. Differences in the dominant primary producer at a given location (e.g., phytoplankton versus SAV) can also determine diel variations in dissolved oxygen and associated ecosystem metabolism. Over a one year period we measured hydrodynamic properties, biogeochemical variables (fDOM, turbidity, chlorophyll-a fluorescence, dissolved oxygen), and photosynthetically active radiation (PAR) at multiple locations in Chincoteague Bay, Maryland/Virginia, USA, a shallow back-barrier estuary. We

quantified light attenuation, net ecosystem metabolism, and timescales of variability for several water properties at paired channel-shoal sites along the longitudinal axis of the bay. The channelized sites, which were dominated by fine bed sediment, exhibited slightly higher light attenuation due to increased wind-wave sediment resuspension. Light attenuation due to fDOM was slightly higher in the northern portion of the bay, while attenuation due to chlorophyll-a was only relevant at one channelized site, proximal to nutrient and freshwater loading. Gross primary

production and respiration were highest at the vegetated shoal sites, though enhanced production and respiration were also observed at one channelized, nutrient-enriched site. Production and respiration were nearly balanced throughout the year at all sites, but there was a tendency for net autotrophy at shoal sites, especially during periods of high SAV biomass. Shoal sites, where SAV was present, demonstrated a reduction in gross primary production (GPP) when light attenuation was highest, but GPP at adjacent shoal sites where phytoplankton were dominant was

less sensitive to light attenuation. This study demonstrates how extensive continuous physical and biological



measurements can help determine metabolic properties in a shallow estuary, including differences in metabolism
and oxygen variability between SAV and phytoplankton-dominated habitats.

**1 Introduction**

Back-barrier estuaries are biologically productive environments that provide numerous ecological, recreational, and
economic benefits. Submerged aquatic vegetation proliferate in these environments due to relatively shallow
bathymetry and a suitable light climate, providing habitat for many fish and crustaceans (Heck and Orth 1980) as
well as enhancing wave attenuation (Nowacki et al. 2017). Primary production in back-barrier estuaries and similar
shallow marine ecosystems is relatively high given the shallow bathymetry, benthic light availability, and sometimes

large SAV beds (e.g., Duarte and Chiscano 1999). Benthic communities within shallow ecosystems host other
primary producers where SAV are absent, including microphytobenthos (Sundbäck et al. 2000) and various forms of
macroalgae, and the relative contribution of these producers is altered by nutrient enrichment (e.g., McGlathery
2001, Valiela et al. 1997).  In deeper, un-vegetated habitats, phytoplankton may also contribute significantly to
primary production, where the balance between water column and benthic primary production is dependent on

depth, the light climate, and nutrient levels, but it is unclear if total ecosystem primary production is affected by
these factors (Borum and Sand-Jensen 1996).

A fundamental control on estuarine primary production is the light climate, which is affected by bathymetry for
benthic primary producers, but is also a function of other spatiotemporally dynamic variables. Models of light
attenuation consider the role of suspended sediment, phytoplankton, and colored dissolved organic matter (CDOM)

concentrations in the water column, either through empirical formulations (Xu et al. 2005) or detailed models of
scattering and absorption properties (Gallegos et al. 1990). Suspended-sediment concentrations are controlled by
processes that vary on a variety of time scales (minutes, weeks, months), including bed composition, bed shear
stress, resuspension, and advective inputs of sediment from external sources. In contrast, phytoplankton
concentrations are a function of light and water column nutrients, while CDOM is driven by the input of terrestrial

material through freshwater loading. The relative contributions of these constituents to the light climate is dependent
on local conditions, can vary spatially and seasonally based on external forcings, and thus requires high-frequency
measurements over space and time to measure all aspects of variability.



A wealth of literature describes the relationship between light and photosynthesis for marine photoautotrophs, including for SAV the role of self-shading, overall water-column conditions, and light attenuation (Sand-Jensen et al. 2006, Kemp et al. 2004, Duarte 1991). Light-photosynthesis interactions for SAV and phytoplankton can differ substantially, given that phytoplankton are vulnerable to water-column structure and mixing while SAV are rooted

to sediments, SAV are vulnerable to epiphytic fouling under nutrient-enriched conditions (Neckles et al. 1993), and SAV can engineer their own light environment through attenuation of flow velocity, wave energy, and therefore bed shear stress and resuspension (Hansen and Reidenbach 2013). Multiple studies have revealed self-reinforcing feedbacks within SAV beds, where SAV shoots and roots stabilize the sediment bed to reduce sediment resuspension, increasing the local net deposition of water-column particulates and improving local light conditions

(e.g., Gurbisz et al. 2017). These feedbacks are complex, however, and depend on bed size, density, and other factors (e.g., Adams et al. 2016). Consequently, healthy SAV beds are likely to generate high rates of primary production relative to adjacent unvegetated areas, but each habitat may respond differently to tidal, diurnal, and event-scale variations in physical forcing.

The purpose of this study is to quantify sub-hourly variations in light climate, water-column properties, and net

ecosystem metabolism and examine the relationship among these properties across habitats in a back-barrier estuary, Chincoteague Bay (Maryland/Virginia, USA) using a year-long deployment of high-frequency sensors. We first describe the observational campaign and analytical methods, followed by analysis of the light climate and associated forcing mechanisms. We then quantify gross primary production, respiration, and net ecosystem metabolism, which have not been studied comprehensively in this estuary, and relate it with the variability of the light climate across

different habitats. Given the large spatial variability in bed sediment type, bathymetry, and dominant vegetation, we aim to quantify how temporal variations in light-attenuating substances, wave dynamics, dissolved oxygen, and metabolic rates are linked across these spatially distinct environments. Our conclusions highlight the importance of quantifying spatiotemporal variability in these processes, which indicate feedbacks between physical and ecological processes in marine environments that should be considered when evaluating future ecosystem response.



## 2 Methods

### 2.1 Site description

Chincoteague Bay, a back-barrier estuary on the Maryland/Virginia Atlantic coast (Fig. 1), spans 60 km from Ocean
City Inlet at the north to Chincoteague Inlet in the south. A relatively undeveloped barrier island separates the
estuary from the Atlantic Ocean. The mean depth is 1.6 m, with depths exceeding 5 m in the inlets. The central basin
depths are approximately 3 m, and the eastern, back-barrier side of the bay is characterized by shallower vegetated
shoals; the western side is deeper with no shoals (Fig. 1).

The coastal ocean tide range approaches 1 m, but is attenuated to less than 10 cm in the center of the bay, where
water levels are dominated by wind setup and remote offshore forcing (Pritchard, 1960). River and watershed
constituent inputs are minimal with the highest discharge and lowest salinities near Newport Bay in the northwest
corner of the bay. Atmospheric forcing is characterized by episodic frontal passages in winter with strong northeast
winds; summer and fall exhibit gentler southwest winds. Waves within the bay are predominantly locally generated,
with substantial dependence on wind direction and fetch due to the alignment of the estuary along a southwest to
northeast axis.

Wazniak et al. (2007) synthesized water quality metrics for Chincoteague Bay, indicating high spatial variability in
chlorophyll-a, dissolved oxygen, and nutrient concentrations. Sites in the northern portion of the bay had generally
poorer water quality, ostensibly due to higher nutrient loading and poorer flushing; Newport Bay had the lowest
water quality index. Fertig et al. (2013) identified terrestrial sources of nutrients to central Chincoteague Bay,
though the precise source (anthropogenic vs. naturally occurring) could not be determined.

### 2.2 Time-series of turbidity, chlorophyll-a, fDOM, dissolved oxygen, and light attenuation

Instrumentation was deployed from 10 August 2014 to 12 July 2015 in order to observe an entire year of forcing
conditions (Suttles et al., 2017) at four locations in Chincoteague Bay (Fig. 1). Instruments were recovered,
downloaded, and serviced three times during that period (October 2014, January 2015, and April 2015). Beginning
in the southern portion of the estuary, site CB03 is within a seagrass meadow (primarily *Zostera marina*) on the
eastern edge of the southern basin, with an above-ground SAV biomass of 61.52 (±9.46) g C m$^{-2}$. Moving
northward, site CB06 is within the main channel north of the nominal boundary between the northern and southern





basins, with a mud-dominated bed and no vegetation. Site CB10 is within a seagrass meadow on the eastern side of the northern basin, with an above-ground SAV biomass of 34.84 ($\pm$3.05) g C m$^{-2}$. Lastly, site CB11 is in a mud-dominated embayment (Newport Bay) within the northwest portion of the estuary. Prior work has identified this region as relatively impaired (i.e. nutrient-enriched) compared to the rest of the estuary (Wazniak et al., 2007). A

meteorological station was deployed at site CBWS, approximately 3.8 m above the water surface at Public Landing, Maryland on the central, western shore of Chincoteague Bay.

The shallow-water platform described by Ganju et al. (2014) was deployed at sites CB03 and CB10, within sandy patches of the seagrass meadows. The platform was designed to measure hydrodynamic, biogeochemical, and light parameters in the bottom half of a 1 m water column, and consists of an RBR D|Wave recorder, a pair of WetLabs

ECO-PARSB self-wiping photosynthetically active radiation (PAR, 400-700 nm) sensors; a YSI EXO multi-parameter sonde measuring temperature, salinity, turbidity, dissolved oxygen, chlorophyll-a fluorescence, fluorescing dissolved organic matter (fDOM, a proxy for CDOM), pH, and depth; and a Nortek Aquadopp ADCP measuring water velocity profiles (2 MHz standard at CB10 and 1 MHz high resolution at CB03). All instruments except for the upper PAR sensor were mounted at 0.15 meters above the bed (mab) on a weighted fiberglass grate

approximately 1 m x 0.5 m. The lower PAR sensor was recessed inside a PVC tube protruding from the bottom of the frame. The upper PAR sensor was mounted at 0.45 mab; the upper and lower sensors provide an estimate of light attenuation $K_{dPAR}$ over the PAR spectrum (400-700 nm), calculated as:

$$K_{dPAR} = -\frac{1}{dz}\ln(PAR_{lower}/PAR_{upper}) \tag{1}$$

where $dz$ is the distance between the two PAR sensors (0.3 m in this case). Light attenuation was calculated only

between the hours of 1030 and 1530, when the angle of the sun relative to the deployment location was closest to 0 degrees. All sensors sampled at 15 min intervals, except for the wave recorders, which burst-sampled at 6 Hz every 3 min. Temperature, turbidity, and inner filter effects (IFE) have been shown to alter fDOM measurements (Baker, 2005; Downing et al., 2012). We corrected fDOM measurements to account for temperature, turbidity, and IFE, according to Downing et al. (2012). Measurements of fDOM at turbidities > 50 NTU were removed due to

interference with the fluorescence signal. Chlorophyll-a concentration was calculated from sensor-based fluorescence measurements by regressing fluorescence against discrete measurements of chlorophyll-a made on four dates (August and October 2014, January and April 2015) at all stations (Fig. S1). In short, water was collected in





the field at the time of sensor sampling, and filtered through 0.7 μm GF/F filters, which were wrapped in foil, and

frozen until laboratory analysis using standard methods (EPA Method 445.0). Non-photochemical quenching (NPQ)

was accounted for by removing fluorescence measurements during periods of peak daylight, and interpolating to fill

gaps. Platforms at sites CB06 and CB11 were identical to platforms at CB03 and CB10 except for the omission of

PAR sensors. Light attenuation at those sites was estimated using the model of Gallegos et al. (2011), discussed

below. Spectral density estimates for dissolved oxygen, turbidity, chlorophyll-a, and fDOM were made using the

WAFO toolbox (Brodtkorb et al., 2000).

Further details on collection protocols and access to the time-series data are reported by Suttles et al. (2017). The

hydrodynamic results of this field campaign have been described in detail by prior studies (Ganju et al. 2016;

Beudin et al. 2017; Nowacki and Ganju 2018); for the purposes of this paper we focus on the spatiotemporal

variability of constituent concentrations and light attenuation.

### 2.3 Estimation of light attenuation contributions

We estimated light attenuation (for periods with missing PAR data or at sites with no PAR data) and relative

contributions from turbidity, chlorophyll-a, and CDOM using the method of Gallegos et al. (2011). This formulation

computes spectral attenuation in terms of suspended and dissolved constituents including the effects of water,

CDOM, phytoplankton, and non-algal particulates (NAP, e.g., detritus, minerals, bacteria). We include absorption

by four components: (1) absorption by water was assumed to follow the spectral characteristics of pure water; (2)

CDOM absorption was taken proportional to fDOM concentration, with a negative spectral slope (Bricaud et al.

1981) set to $s_g$=0.02 nm$^{-1}$ (Oestreich et al., 2016); (3) phytoplankton absorption was proportional to chlorophyll $a$

concentration and with the spectrum shape normalized by the absorption peak at 675 nm (initial value for peak

absorption was taken as $a_{\psi,675}$=0.0235 m$^2$ (mg chl $a$)$^{-1}$, within the range provided by Bricaud et al. 1995); and (4)

non-algal absorption was taken as proportional to the suspended-sediment concentration with a spectral shape

(Bowers and Binding, 2006) that included a baseline of $c_{x1}$=0.0024 m$^2$ g$^{-1}$ (Biber et al. 2008), an absorption cross-

section of $c_{x2}$=0.04 m$^2$ g$^{-1}$ (Bowers and Binding 2006), and a spectral slope of $s_x$=0.009 (Boss et al. 2001). The

backscattering ratio of water was set at 0.5, while CDOM is considered non-scattering (Mobley and Stramski 1997),

and the particulate effective backscattering ratio $b_{bx}$ was initially set at 0.017.




Given the high variability in turbidity and suspended-sediment concentrations due to wind-wave resuspension, we varied the value of $b_{bx}$ as a function of turbidity to achieve the best agreement between the model and observations. At turbidity below 50 NTU, $b_{bx}$ is constant at 0.017, and then linearly declines to 0.0024 at turbidities above 220 NTU. This modification implies that backscattering by mineral particles dominates at low turbidities, while large

organic aggregates dominate at the highest turbidities (Gallegos et al. 2011), and has the effect of enhancing attenuation by particulate-induced absorption at the highest turbidities. The lowest values of backscattering to scattering ratio $b_{bx}$ in the literature range from 0.005 (Snyder et al. 2008) to about 0.002 (Chang et al. 2004). The chosen value of 0.0024 was obtained from Loisel et al. (2007). Morel and Bricaud (1981) described the $b_{bx}$ ratio as decreasing with increasing absorption, which would be consistent with high turbidity situations. McKee et al. (2009)

described a decrease of the $b_{bx}$ ratio with increasing concentrations in a mineral-rich environment. The maximum sediment concentrations (15 mg/l) in that study were higher than previously mentioned studies, but smaller than concentrations observed in this environment (Nowacki and Ganju 2018). Determining the appropriate backscattering ratio at high turbidities is still poorly constrained and the minimum value used in this study might even be an overestimation.

**2.4 Net ecosystem metabolism**

The basic concept and method for computing community production and respiration (and ecosystem metabolism) was developed by Odum and Hoskin (1958) and, with numerous modifications, has been used since for estimating these rate processes in streams, rivers, lakes, estuaries and the open ocean (Caffrey 2004; Howarth et al. 2014). The technique is based on quantifying increases in oxygen concentrations during daylight hours and declines during

nighttime hours as ecosystem rates of net primary production and respiration, respectively. The sum of these two processes over 24 h, after correcting for air-sea exchange, provides an estimate of net ecosystem metabolism. We utilized continuous oxygen concentration measurements at four locations in Chincoteague Bay (CB03, CB10, CB06, CB11) to compare differences in net ecosystem metabolism across sites across habitats (phytoplankton versus SAV dominated, channel versus shoal). We computed daily estimates of gross primary production ($P_g$), ecosystem

respiration ($R_t$), and net ecosystem metabolism ($P_n = P_g - R_t$) using the approach of Beck et al. (2015), which utilizes weighted regression to remove tidal effects on dissolved oxygen time-series. The calculations utilized salinity, temperature, and dissolved oxygen times-series from the YSI EXO sensors and water level at each platform, and





atmospheric conditions from the weather station (for wind speed) and a nearby buoy (OCIM2 - 8570283 at the

Ocean City Inlet, Maryland) for atmospheric pressure and air temperature.

**2.5 SAV biomass**

In August 2014 and April 2015, we estimated SAV biomass by collecting triplicate cores of above-ground and

below ground biomass using 150 cm$^2$ plexiglass cores. Plant material was separated from sediments and the living

and non-living plant material was sorted, rinsed with DI water, and dried in an oven until repeated weight

measurements were constant. The dried material was ground using mortar and pestle and analyzed for carbon and

nitrogen content using elemental analysis (USEPA 1997). These biomass measurements (Table 2) were made at

CB10 and CBO3 where SAV were present.

**3 Results**

**3.1 Time-series of turbidity, chlorophyll-a, fDOM, dissolved oxygen, and light attenuation**

Turbidity ranged from near zero to a maximum of over 400 NTU at site CB06 during a winter storm that induced

waves exceeding 0.7 m (Figs. 2-5). The highest turbidities were observed at CB06, while sites CB03, CB10, and

CB11 had similar statistical distributions of turbidity. In general, tidal resuspension appeared to be minimal although

tidal advection after large wind-wave resuspension events was observed (Figs. 2-5). Spectrally, most of the energy

in the turbidity signal was found at subtidal frequencies (i.e., > 1 day), with little correspondence between sites in

frequency space (Fig. 6).

Chlorophyll-a concentrations peaked just below 50 µg/L at site CB11, and below 30 µg/L at all other sites (Figs. 2-

5). Chlorophyll-a concentrations were comparable across sites CB03, CB10, and CB06, but on average twice as high

at CB11 (Table 2). The largest concentrations were observed in winter during resuspension events and during a

broad spring bloom during March-April 2015. All sites showed a large decrease in chlorophyll-a concentration

during a period of ice formation in late February 2015, when advection and resuspension were largely halted in the

estuary. Despite removing effects of NPQ, an attenuated diel signal remains in the spectra that may be a partial

artifact or consistent with day-night variations in PAR (Fig. 6). The diurnal signal was comparable across all stations

and was stronger than the tidal signal (Fig. 6).





Concentrations of fluorescing dissolved organic matter (fDOM) varied between 0 and 70 QSU, with the highest values consistently observed at CB11 in Newport Bay (Figs. 2-5). Concentrations were similar between the other sites, with lowest fDOM in the winter, possibly due to reduced biological activity. Periodic large decreases in fDOM coincided with increases in turbidity due to wind-wave resuspension; despite correcting the fDOM time-series for

turbidity interference, the fluorescence measurement was likely attenuated beyond correction.  Most of the energy in the fDOM signal was at subtidal frequencies (Fig. 6), there was a distinct peak at the $M_2$ tidal frequency (12.42 h) corresponding to advection of fresher, fDOM-elevated water on ebb tides.

Dissolved oxygen percent saturation ranged from a minimum of 20% at site CB11 in the summer, to a maximum of over 200% at site CB10 (Fig. 7), with maximum diel variability in the summer. The channel sites (CB06 and CB11)

showed substantially attenuated diel variations as compared to the shoal sites (CB03 and CB10). Diel fluctuations in the winter were typically less than the subtidal changes. Spectral analysis clearly shows the dominance of 24 h diel fluctuations at all sites (Fig. 6), with higher energy at the vegetated shoal sites (CB03, CB10).

Direct measurements of $K_{dPAR}$ at sites CB03 and CB10 were partially confounded by instrument fouling and malfunction of one or both sensors on each platform. Nonetheless, we successfully captured a wide range of

conditions, with peak light attenuation of approximately 10 m$^{-1}$ occurring at sites CB03 and CB10 in the winter during a sediment resuspension event (Figs. 2-5). Median $K_{dPAR}$ was approximately 1 m$^{-1}$ at both shoal sites, but event-driven magnitudes exceeded 2 m$^{-1}$ multiple times during the deployment. The $K_{dPAR}$ data were used to calibrate the light model (Fig. S2), which we implemented to reconstruct missing data at sites CB03 and CB10, and to estimate light attenuation at sites CB06 and CB11 where constituent concentrations were measured but no light

data were collected. The full time-series of estimated light attenuation at the four sites demonstrates the strong seasonal variability of light attenuation at all sites (Figs. 2-5), with peak attenuation occurring during winter storms. Wave-induced sediment resuspension and advection were responsible for increased turbidity, which accounted for approximately 40% of the light attenuation at sites CB03, CB10, and CB11, and 61% at site CB06. Light attenuation at site CB11, with its proximity to freshwater and nutrient sources, was highest overall and more highly influenced

by chlorophyll-a and fDOM than at other sites. Median $K_{dPAR}$ was highest at the two unvegetated channel sites, and lowest at the two vegetated shoal sites (Table 1). Turbidity, the strongest driver of $K_{dPAR}$, was generally lower





during the warm season at the vegetated shoal sites and the turbidity generated for a given wind-wave induced bed

shear stress was substantially reduced in summer at the vegetated sites (Fig. 8).

**3.2 Net ecosystem metabolism**

Estimates of ecosystem metabolism displayed strong seasonal variability, with elevated rates of $P_g$ and $R_t$ during

warm months across all stations (Figs. 9, 10). High rates of $P_g$ and $R_t$ persisted between May and October, when

temperature and PAR peaked seasonally, and were consistently lower during November to April (Figs. 9, 10), and $R_t$

was exponentially related to temperature across sites (although less so at CB06). Metabolic rates were clearly higher

at vegetated shoal sites (CB03, CB10), consistent with the strong diurnal signal in oxygen at these sites (Figs. 6, 7)

and the presence of seagrass (Table 2). Temperature was strongly associated with rates of respiration across all sites,

and respiration reached peak values under conditions of high temperatures and high rates of GPP (Fig. 10). Gross

primary production and respiration were largely balanced across all sites, but instances of net autotrophy ($P_g > R_t$)

occurred nearly 70% of the time at the vegetated sites, and were also persistent at CB06 (Fig. 11). $P_g$ and $R_t$ were

more balanced at CB11, but we did not have enough data to capture the complete seasonal cycle at this site (Fig. 9).

The relationship between rates of $P_g$ and light availability was site specific. At the vegetated sites, CB03 and CB10,

$P_g$ and $K_{dPAR}$ were negatively correlated, with high rates of $P_g$ occurring during the periods of lowest $K_{dPAR}$ and

highest surface PAR (Fig. 11). At CB06 and CB11, variations in $P_g$ and $K_{dPAR}$ were not strongly related and $K_{dPAR}$

was higher and $P_g$ was lower at these sites than at CB03 and CB10 (Table 1,2; Fig. 11, 12). The highest $P_g$ values

did correspond to days with the highest surface PAR at CB06 and CB11. In summary, the highest metabolic rates we

measured occurred during warm periods in vegetated shoals, where wind-wave attenuation reduced turbidity and

$K_{dPAR}$.

**4 Discussion**

An analysis of a comprehensive suite of high-frequency biological and physical measurements in Chincoteague Bay

over an annual cycle revealed the primary drivers of light attenuation, the role of light attenuation in driving

variations in gross primary production, the primary timescales of biogeochemical variability, and the effect of

habitat type (i.e., vegetated versus un-vegetated; nutrient enrichment) on oxygen variability and net ecosystem





metabolism. Turbidity dominated light attenuation variability and varied considerably at 1-7 day time scales, consistent with the frequency of storm passage. Storm-associated wind waves were the specific driver of resuspension and turbidity, and reduction of bed shear stress and turbidity in SAV-dominated shoal environments during summer increased light availability in these habitats. As a consequence, GPP was substantially higher in

SAV-dominated shoals compared with adjacent plankton-dominated sites and GPP was negatively correlated with $K_{dPAR}$ in the shoals, highlighting the role of short-term variability in light availability driving GPP. High rates of GPP and R in shoal environments led to much higher diurnal and seasonal--scale variability in dissolved oxygen in these habitats.

**4.1 Spatiotemporal variability of light attenuation**

Light attenuation can appear to be controlled by uncoupled biological (i.e., nutrient loading and phytoplankton blooms) and physical (sediment resuspension) processes, but in reality the feedbacks between physics and biology are consistently present in estuaries. Wave-induced suspended-sediment resuspension is primarily responsible for light attenuation in shallow lagoons like Chincoteague Bay, and seagrass clearly modulates the magnitude and spatiotemporal variability in sediment resuspension through the attenuation of wave energy (e.g., Hansen and

Reidenbach 2013). This wave-induced resuspension and turbidity occurs on the time scale of periodic wind events observed in this system. During summer months, when vegetation densities are the highest, the dependence of turbidity on bed shear stress is highest at the unvegetated channel sites CB06 and CB11, and lowest at the vegetated shoal sites CB03 and CB10 (Fig. 8). In the winter, when vegetation densities are lowest, the dependence increases at the shoal sites, indicating the influence of seagrass on bed stabilization in the summer. The role of seagrass in

modulating their physical environment is demonstrated by this improvement in light climate when vegetation re-establishes in the warmer months.

Our measurements underscore the ability of comprehensive continuous measurements to capture multiple scales of variation in light attenuation. For example, subsampling our light attenuation measurements under fair weather conditions (wave height < median) leads to underestimation of median $K_{dPAR}$ by as much as 27% (sites CB06 and

CB10). Subsampling to periods with wave heights less than the 84[th] percentile leads to underestimation by 13% (site CB06). The effect of these changes can be large if sparse data are used to assess trends in water quality, or to drive ecological models. In situations where continuous monitoring of light attenuation is difficult, light modeling using



continuous measurements of turbidity, chlorophyll-a, and fDOM is a suitable proxy. With a reasonable number of

discrete light attenuation samples, it is possible to calibrate a light model and estimate seasonal changes in

contributions to light attenuation using the constituent measurements. This becomes useful when attempting to

determine the relative influence of physical and biogeochemical processes on spatiotemporal variations in light

climate, and the potential benefits of restoration activities.

Apart from Newport Bay in the northwest corner of the estuary, Chincoteague Bay is relatively unimpacted by

external nutrient and organic material inputs, as compared to other lagoonal systems on the U.S. East Coast (e.g.

Indian River Lagoon, Phlips et al. 2002; Great South Bay, Kinney and Valiela 2011). For example, the relatively

urbanized watershed surrounding Barnegat Bay, New Jersey, contributes larger nutrient loads to the northern portion

of the estuary (Kennish et al. 2007), however the highest light attenuation is observed in the southern portion, where

sediment concentration are highest due to the availability of fine bed sediment and wind-wave resuspension (Ganju

et al. 2014). In fact, in Barnegat Bay light attenuation is lowest in the more eutrophic, poorly flushed (Defne and

Ganju 2015) southern portion due to coarser bed sediments and smaller wave heights (and less resuspension). In the

more nutrient-enriched regions of the Maryland coastal bays north of Chincoteague Bay, chlorophyll-a is also much

higher and likely makes a larger contribution to light attenuation (Boynton et al. 1996). In contrast, spatial gradients

in the light field within Chincoteague Bay are driven by geomorphology (depth), the presence of submerged aquatic

vegetation (CB03 and CB10), and the higher contribution of chlorophyll-a at a nutrient-enriched site (CB11;

Boynton et al. 1996). Thus, generalizing the light climate in back-barrier estuaries is hampered by these subtle,

habitat-specific controls on attenuation, but improvements in continuous monitoring will lead to an increased

understanding of these controls.

**4.2 Spatiotemporal variability in metabolism and relationship with benthic ecosystem and light climate**

Clear differences in the magnitude and timescales of oxygen dynamics were present across habitats in Chincoteague

Bay. Spectral density at the diurnal time-scale for dissolved oxygen was much higher at the two vegetated sites,

where high rates of primary production during the day and associated respiration at night led to large changes in

dissolved oxygen. Although this pattern is unsurprising given the high metabolic rates in these dense seagrass beds,

this study is one of few studies that have clearly documented this pattern inside and outside of SAV beds at adjacent

sites. Elevated metabolic rates in macrophyte dominated habitats relative to phytoplankton dominated habitats have

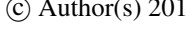



been documented in other systems (D'Avanzo et al. 1996), where the C:N molar ratio of phytoplankton (C:N = 6.6) is much less than that measured for Chincoteague Bay *Z. marina* (C:N = 25.8±7.2), indicating higher carbon and oxygen metabolism for a given amount of nitrogen (e.g., Atkinson and Smith 1983). Elevated magnitudes of daily oxygen change have also been associated with elevated phytoplankton biomass in Chincoteague Bay (e.g., Boynton

et al. 1996) and other systems (e.g., D'Avanzo et al. 1996). This is consistent with the fact that the site at Newport Bay (CB11), which had the highest plankton chlorophyll-a concentrations across all sites, had a larger spectral density for dissolved oxygen at the diel frequency than at CB06. Newport Bay is known to have elevated nutrient inputs and concentrations associated with land-use in its watershed (Boynton et al. 1996), and spectral density for chlorophyll-a was highest at this location (Fig. 6), suggesting that this site is responding to eutrophication.

Seasonal variations in metabolic rate estimates were typical of temperate regions, but were variable across space. At all stations, rates of $P_g$ and $R_t$ were highest in warmer months when incident PAR is highest in this region (e.g., Fisher et al. 2003), but rates were especially high in the later summer (August and September) at the vegetated sites. This late-summer period is typically when SAV biomass is highest and where the self-reinforcing effect of wave attenuation and reduced $K_{dPAR}$ allow for high rates of primary production (Fig. 8). Indeed, the relationship between

$K_{dPAR}$ and GPP was strongest at these vegetated sites, displaying the sensitivity to benthic primary production to light availability, which was dominated by turbidity. Elevated temperatures during this period also stimulate respiration (e.g., Marsh et al. 1986), which has been documented at the ecosystem scale in other SAV-dominated coastal systems (Howarth et al. 2014, Boynton et al. 2014). Seasonal variations in metabolic rates were comparatively lower at the unvegetated, non-eutrophic site (CB06), where low phytoplankton biomass and the

absence of macrophytes leads to limited primary production and respiration rates.

The metabolic balance (ratio) between primary production and respiration is a metric of interest in coastal aquatic ecosystems as it provides an indication of whether the system is a relative source or sink of carbon with respect to the atmosphere (Stæhr et al. 2006). Many river-dominated estuaries and estuaries flanked by extensive tidal marshes are net heterotrophic, where respiration exceeds primary production (e.g., Kemp and Testa 2011). $P_g$ and $R_t$ are often

balanced or indicate modest autotrophy in shallow lagoons where SAV is present (e.g., Ferguson and Eyre 2010, D'Avanzo et al. 1996). Autotrophy tends to persist in these habitats as submerged macrophytes like SAV generate high biomass where light availability is relatively high given low nutrient concentrations, low phytoplankton





biomass, and sediment trapping within SAV beds. In fact, modest net autotrophy prevailed during the summer

season at vegetated sites but not at un-vegetated sites (Fig. 7). The rates of $P_g$ and $R_t$ we estimated, which were

typically between 200 and 400 mmol $O_2$ m$^{-2}$ d$^{-1}$, are comparable to those measured in other temperate ecosystems

dominated by *Zostera marina* (Howarth et al. 2014) and net autotrophy in these low-nutrient environments is

consistent with other coastal lagoons in the region (Giordano et al. 2012, Stutes et al. 2007). While the overall

metabolic balance (P/R) of Chincoteague Bay is difficult to assess, given large differences in metabolic balance

across gradients of depth and nutrient enrichment, we made a simple calculation where mean rates of net ecosystem

metabolism at CB10 were multiplied by the area of SAV in 2015 (30 km$^2$) and rates from CB06 are multiplied by

the remaining area without SAV (276 km$^2$). The calculation assumes that rates measured at these stations are

representative of similar habitats across Chincoteague Bay, and does not account for the potential of benthic

photosynthesis to occur in shallow regions not occupied by SAV. This approaches generates an estimate of net

metabolism for Chincoteague Bay of 317±461 Mmol $O_2$ yr$^{-1}$, which indicates autotrophy basin-wide, but the

uncertainty is high.

$P_g$ and $R_t$ were tightly coupled in the Chincoteague Bay stations during 2014-2015, suggesting that primary

producers were the dominant sources of respiration in the ecosystem. At the vegetated stations, low plankton

chlorophyll-a levels indicate that SAV were the dominant contributors to ecosystem metabolism, though macroalgae

and epiphytic algae were also present. Alternatively, high chlorophyll-a levels at CB11 (Newport Bay) suggest that

phytoplankton were dominant contributors to metabolism, as this site is known to be nutrient enriched. Alternate

sources of primary production and respiration that could drive metabolism include benthic micro- and macro-algae

and heterotrophic bacteria. Previously measured sediment oxygen uptake (SOU) rates during summer in sediment

incubations without light suggest a mean SOU of 56.2 (±24.7) mmol $O_2$ m$^{-2}$ d$^{-1}$ (Bailey et al. 2005) which is ~50%

of the mean summer respiration rates at CB06, but only 15% of rates measured at the phytoplankton-dominated

CB11. Rates of benthic photosynthesis were not available at this site, but with a mean $K_{dPAR}$ of 1.35 at CB06 and a

water-column depth of 3 m, only a small fraction of surface PAR would be expected to reach the sediments. Thus,

respiration at CB06 might be evenly split between the water-column and sediments, which fits well with previous

cross-system comparisons of coastal marine ecosystems (Kemp et al. 1992, Boynton et al. 2018).



**5 Conclusions**

We explored the relationship between light climate, net ecosystem metabolism, geomorphology, and benthic community structure at four sites in a back-barrier estuary over one year, finding clear linkages among submerged aquatic vegetation density, sediment resuspension, light attenuation, and ecosystem metabolism. Vegetated shoal

sites exhibited higher metabolic rates, reduced sediment resuspension, and reduced light attenuation as compared to unvegetated channel sites. Light attenuation was dominated by wind-wave induced sediment resuspension, which peaked in winter months when vegetation was at its lowest density. Ecosystem production and respiration were largely balanced across all sites, but tended toward modest net autotrophy at sites vegetated by SAV. These results demonstrate the need for high-temporal resolution measurements at multiple locations within a given estuary, due to

the strong interplay between geomorphology, light climate, seagrass dynamics, and ecosystem metabolism. The mechanisms identified here demonstrate a need to consider feedbacks between biological and physical processes in estuaries, especially when constructing deterministic models or evaluating future ecosystem function.

**Code availability**

Freely available MATLAB toolboxes, as cited in the text, were used for analyses.

**Data availability**

The time-series data are available at https://stellwagen.er.usgs.gov/chincoteague.html

**Sample availability**

N/A

**Appendices**

N/A

**Team list**

N/A




**Author contribution**

N.K.G., S.E.S, and J.M.T. designed the study, A.L.A. implemented the light model, all authors analyzed time-series data and drafted the manuscript.

**Competing interests**

The authors declare that they have no conflict of interest.

**Disclaimer**

Use of brand names is for identification purposes only and does not constitute endorsement by the U.S. Government.

**Acknowledgments**

We thank Alexis Beudin, Dan Nowacki, Sandra Brosnahan, Dave Loewensteiner, Nick Nidzieko, Ellyn

Montgomery, and Pat Dickhudt for field and data assistance. Casey Hodgkins and Amanda Moore provided help with sample processing and data analysis. W. Michael Kemp provided constructive feedback on the manuscript. This study was funded by the USGS Coastal and Marine Geology Program, the Department of the Interior Hurricane Sandy Recovery program (GS2-2D). Time-series data can be accessed at the USGS Oceanographic Time-Series Database at http://dx.doi.org/10.5066/F7DF6PBV. This is UMCES Contribution Number XXXX.

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





**Figure captions**

Figure 1. Location map with bathymetry, instrument locations, and submerged aquatic vegetation coverage from the Virginia Institute of Marine Science annual SAV mapping survey (http://web.vims.edu/bio/sav/). WQ refers to water-quality sonde; PAR refers to paired upper/lower PAR sensors.

Figure 2. Time-series of upper and lower PAR, $K_{dPAR}$, turbidity, chlorophyll-a, and fDOM at vegetated shoal site CB03.

Figure 3. Time-series of upper and lower PAR, $K_{dPAR}$, turbidity, chlorophyll-a, and fDOM at vegetated shoal site CB10.

Figure 4. Time-series of modeled $K_{dPAR}$, turbidity, chlorophyll-a, and fDOM at unvegetated channel site CB06.

Figure 5. Time-series of modeled $K_{dPAR}$, turbidity, chlorophyll-a, and fDOM at unvegetated channel site CB11.

Figure 6. Spectral density estimates for dissolved oxygen, turbidity, chlorophyll-a, and fDOM at all four sites.

Figure 7. Time-series of dissolved oxygen from four sites.

Figure 8. Relationship between combined wave-current induced bed shear stress and turbidity at four sites, with linear regressions of bin-averaged values over summer (May-September) and winter (October-April) seasons.

Figure 9. Monthly estimates of gross primary production ($P_g$), respiration ($R_t$), and net ecosystem metabolism (NEM) at four sites. Measurements span August 2014 through July 2015, therefore time axis begins in August 2014 and wraps back to January 2015.

Figure 10. Relationship between respiration and water temperature at four sites, with data coloration scaled to gross primary production.

Figure 11. Relationship between gross primary production and respiration at four sites, line of 1:1 agreement shown. %of estimates indicating autotrophy or heterotrophy are indicated.

Figure 12. Relationship between light attenuation $K_{dPAR}$ and gross primary production. Significant reduction in gross primary production at the highest $K_{dPAR}$ is observed at vegetated shoal sites.





**Tables**

20    Table 1. Median light attenuation and relative contributions from turbidity, chlorophyll-a, and fDOM. Remainder of

light attenuation contribution is from water (not shown), and equivalent between sites.





|  | **CB03** | **CB06** | **CB10** | **CB11** |
|---|---|---|---|---|
| $K_{dPAR}$ (m$^{-1}$) | 1.00 | 1.35 | 1.19 | 1.67 |
| $K_{dPAR}$ (turbidity) | 0.44 (44%) | 0.82 (61%) | 0.53 (45%) | 0.66 (40%) |
| $K_{dPAR}$ (chl-a) | 0.11 (11%) | 0.10 (7%) | 0.11 (10%) | 0.37 (22%) |
| $K_{dPAR}$ (fDOM) | 0.17 (17%) | 0.15 (11%) | 0.24 (20%) | 0.31 (19%) |

Table 2. Water-column properties at the four study sites, including SAV biomass (where present) and metabolic rate

estimates. Temperature data are minimum and maximum values (min, max), salinity is the annual mean,

chlorophyll-a is the annual mean (with maximum in parentheses), SAV biomass is the mean above-ground biomass

5   (±*SD*), and metabolic rates estimates are means (±*SD*) for the August-September (peak SAV growth) period.

|  | **CB03** | **CB06** | **CB10** | **CB11** |
|---|---|---|---|---|
| *Depth (m$^{-1}$)* | 1 | 3 | 1 | 2.5 |
| *Temperature (degC)* | -1.64 – 30.81 | -1.62 – 29.06 | -1.66 – 30.10 | -1.51 – 29.15 |
| *Salinity* | 26.79 | 26.67 | 26.15 | 24.52 |
| *Chlorophyll-a (µg/L)* | 5.54 (29.5) | 5.81 (29.9) | 5.89 (19.9) | 11.73 (46.7) |
| *SAV Biomass (g C m$^{-2}$±SD)* | 61.52 (±9.46) | N/A | 34.84 (±3.05) | N/A |
| *Pg (mmol O$_2$ m$^{-2}$ d$^{-1}$)* | 310.7 (±162.2) | 93.2 (±53.7) | 281.9 (±198.5) | 239.9 (±133.9) |
| *Rt (mmol O$_2$ m$^{-2}$ d$^{-1}$)* | 304.1 (±179.7) | 92.9 (±55.3) | 267.0 (±198.2) | 252.2 (±133.4) |
| *Pn (mmol O$_2$ m$^{-2}$ d$^{-1}$)* | 6.7 (±36.8) | 0.3 (±27.1) | 14.9 (±46.6) | -12.3 (±57.8) |

**Figures**



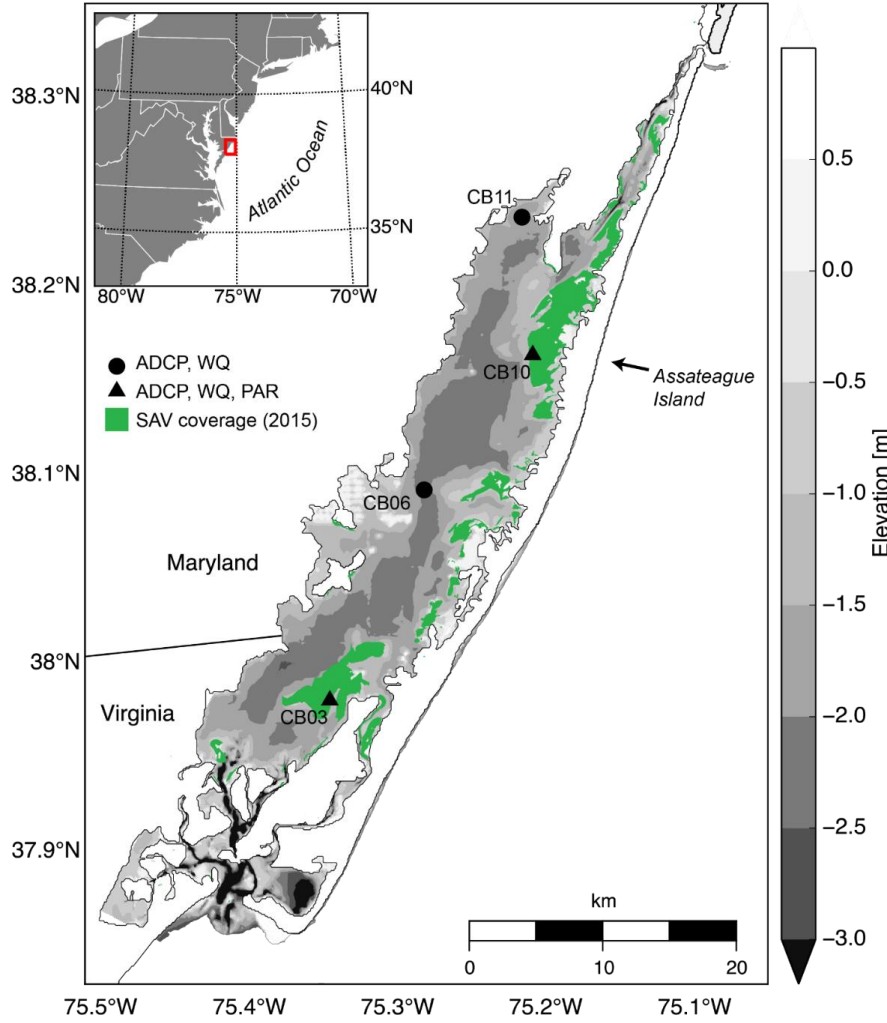

Figure 1. Location map with bathymetry, instrument locations, and submerged aquatic vegetation coverage from the

Virginia Institute of Marine Science annual SAV mapping survey (http://web.vims.edu/bio/sav/). WQ refers to

water-quality sonde; PAR refers to paired upper/lower PAR sensors.




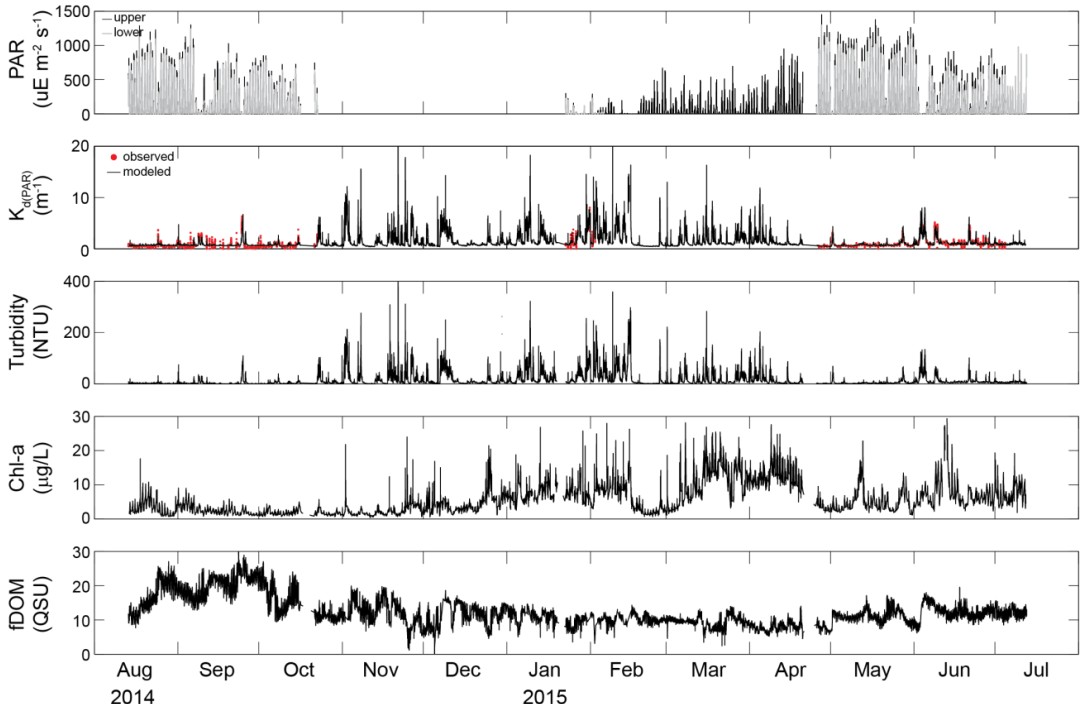

Figure 2. Time-series of upper and lower PAR, $K_{dPAR}$, turbidity, chlorophyll-a, and fDOM at vegetated shoal site

CB03.



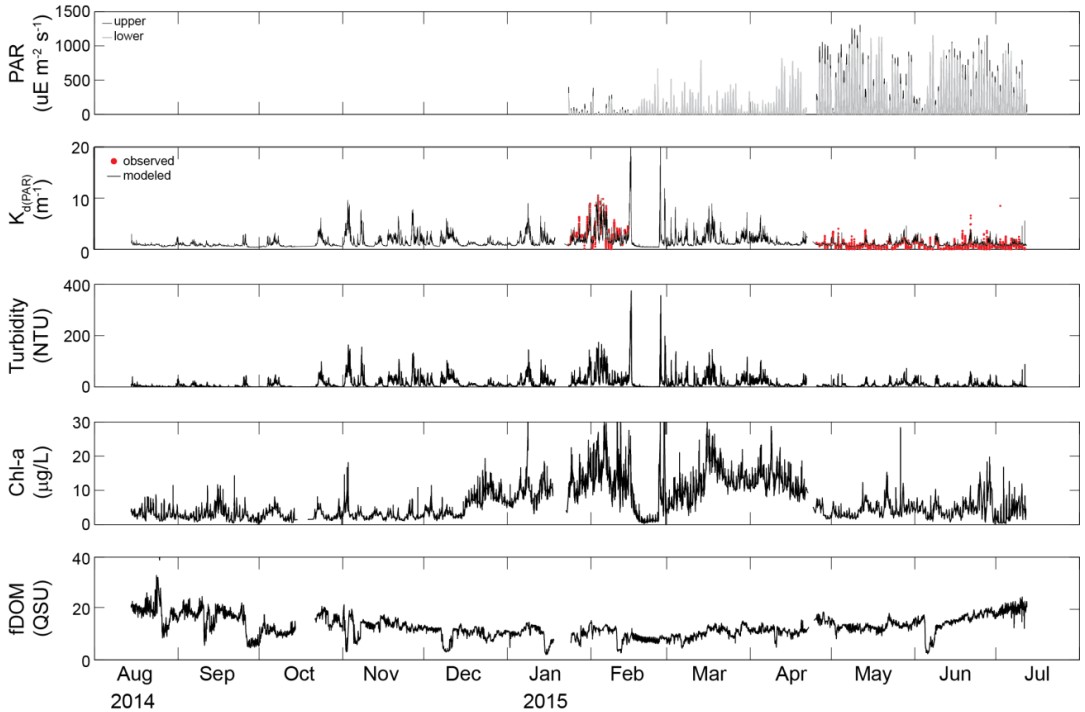

Figure 3. Time-series of upper and lower PAR, $K_{dPAR}$, turbidity, chlorophyll-a, and fDOM at vegetated shoal site

CB10.

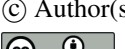



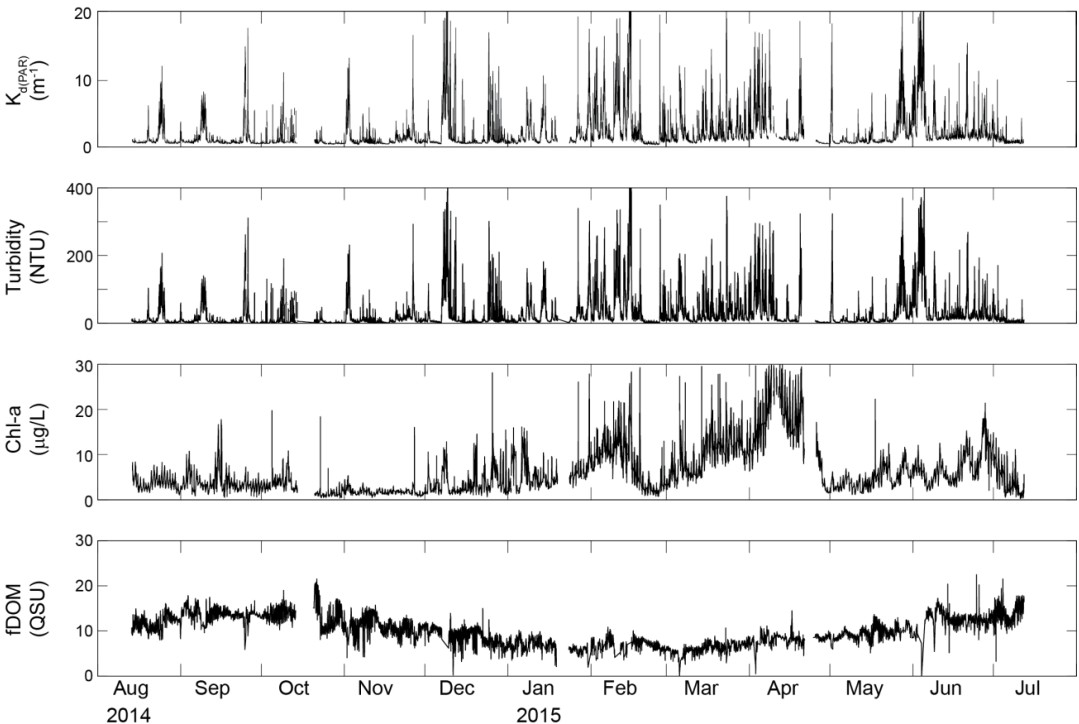

Figure 4. Time-series of modeled $K_{dPAR}$, turbidity, chlorophyll-a, and fDOM at unvegetated channel site CB06.



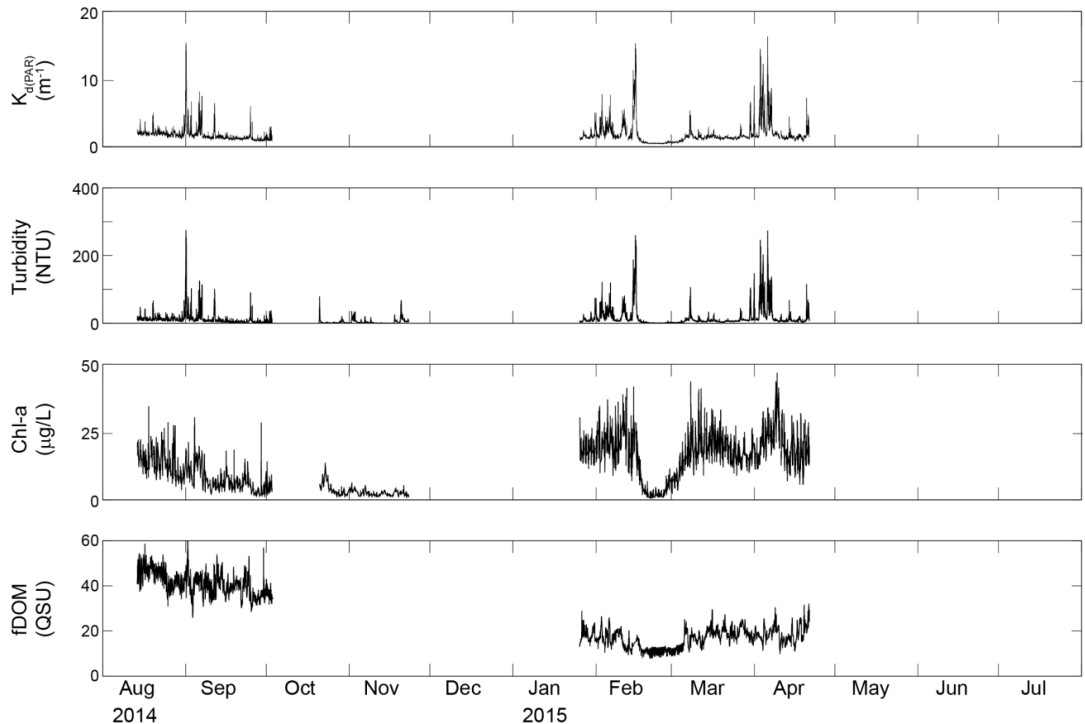

Figure 5. Time-series of modeled $K_{dPAR}$, turbidity, chlorophyll-a, and fDOM at unvegetated channel site CB11.





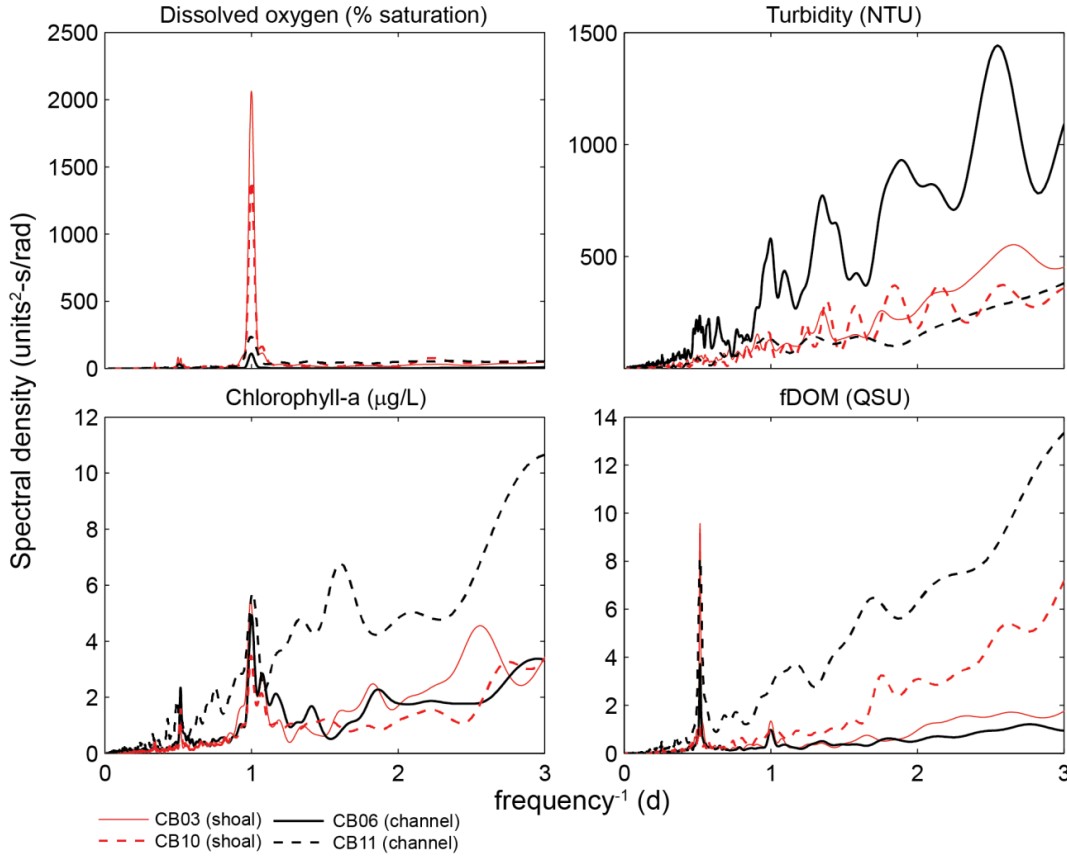

Figure 6. Spectral density estimates for dissolved oxygen, turbidity, chlorophyll-a, and fDOM at all four sites.



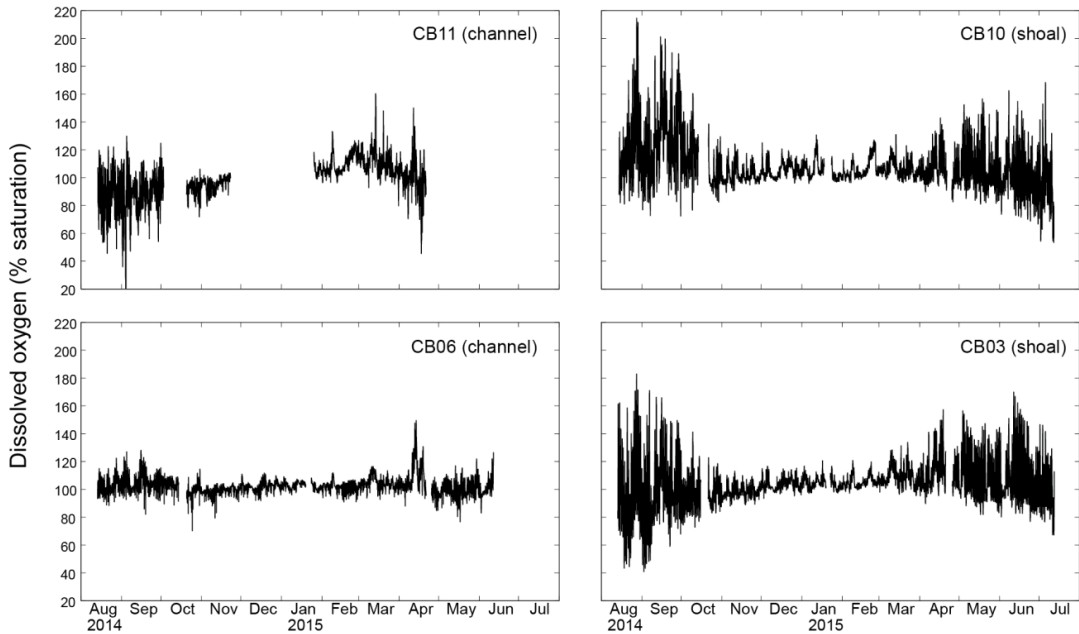

Figure 7. Time-series of dissolved oxygen from four sites.





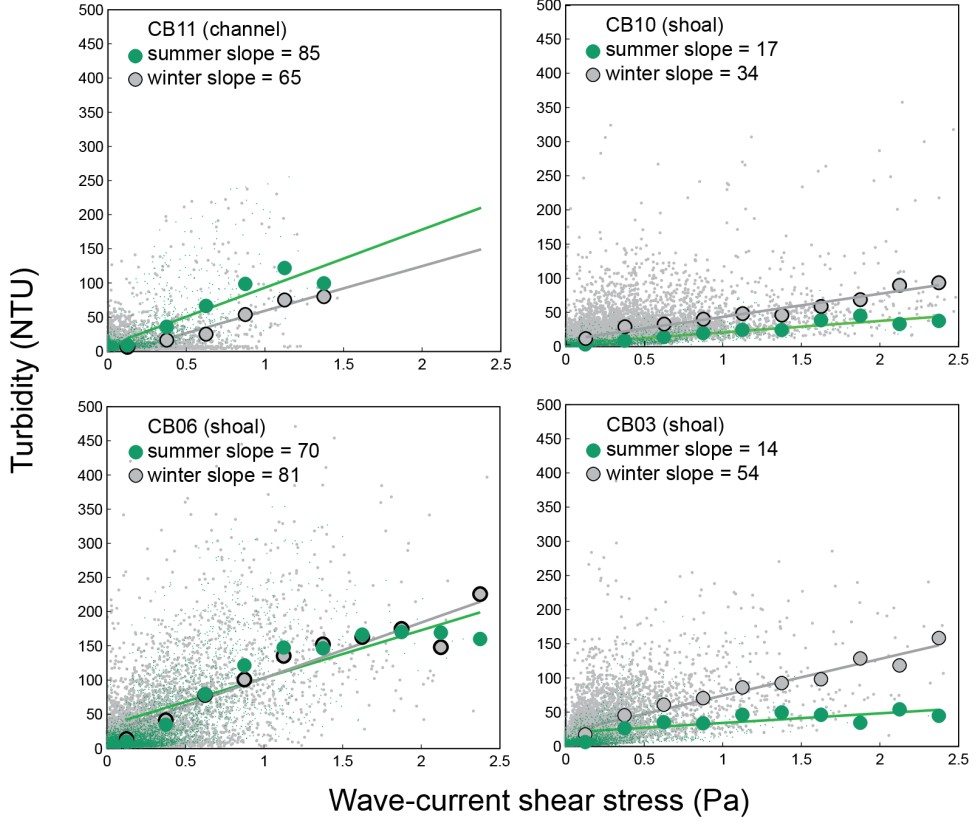

Figure 8. Relationship between combined wave-current induced bed shear stress and turbidity at four sites, with

linear regressions of bin-averaged values over summer (May-September) and winter (October-April) seasons.



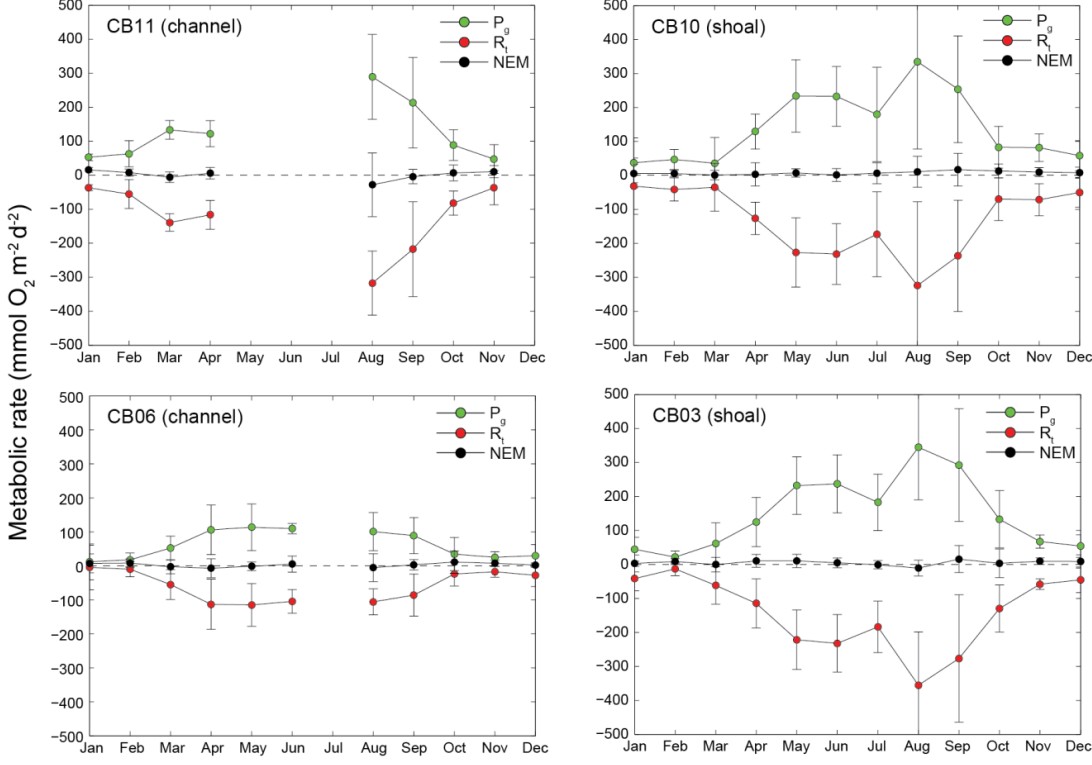

Figure 9. Monthly estimates of gross primary production ($P_g$), respiration ($R_t$), and net ecosystem metabolism

(NEM) at four sites. Measurements span August 2014 through July 2015, therefore time axis begins in August 2014

and wraps back to January 2015.





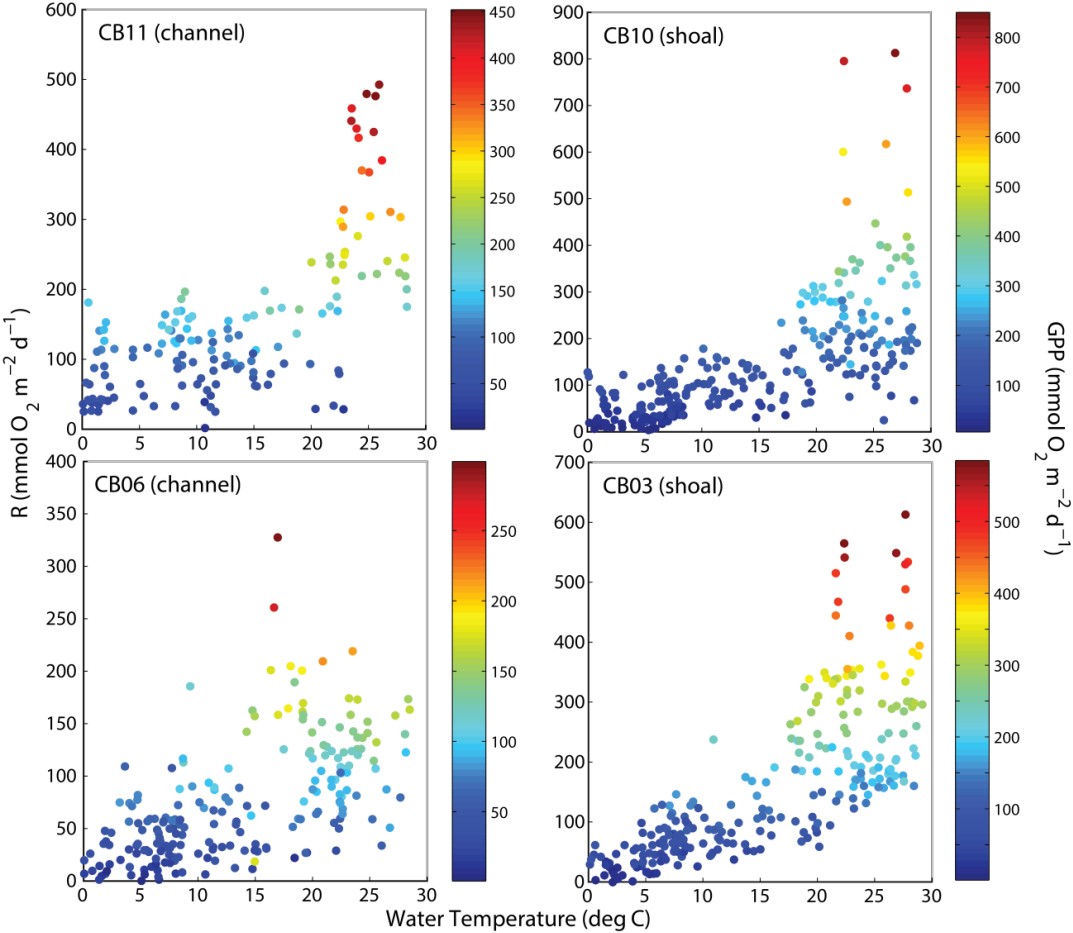

Figure 10. Relationship between respiration and water temperature at four sites, with data coloration scaled to gross primary production.



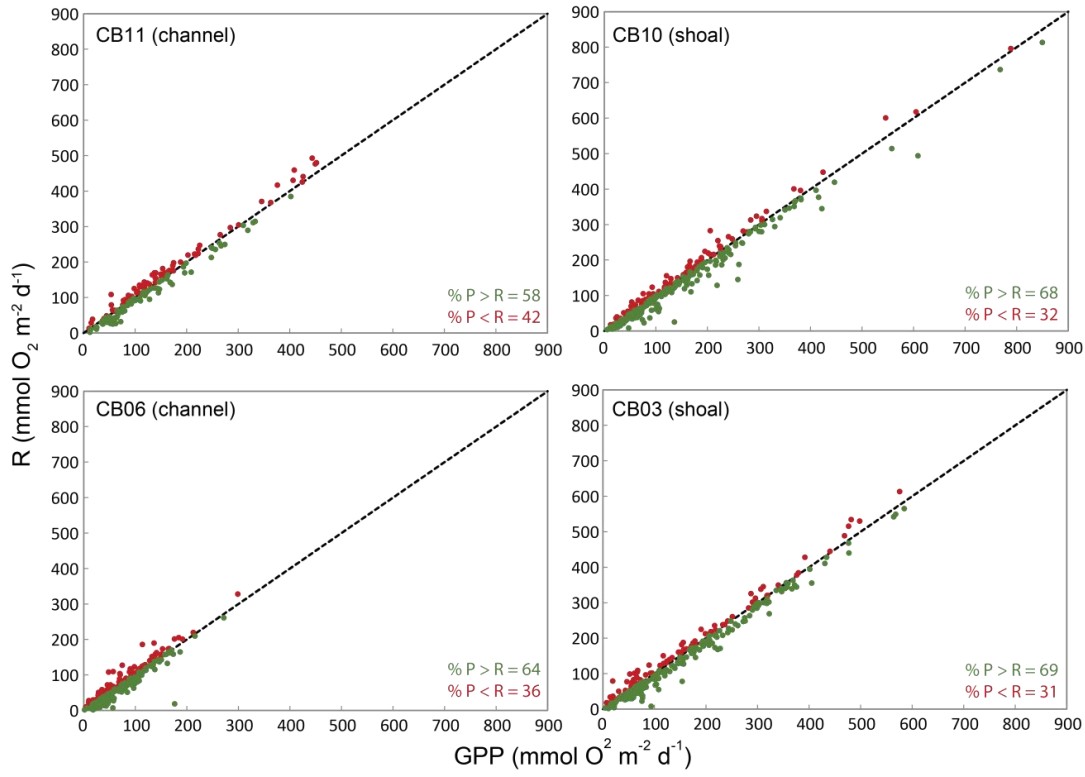

Figure 11. Relationship between gross primary production and respiration at four sites, line of 1:1 agreement shown.



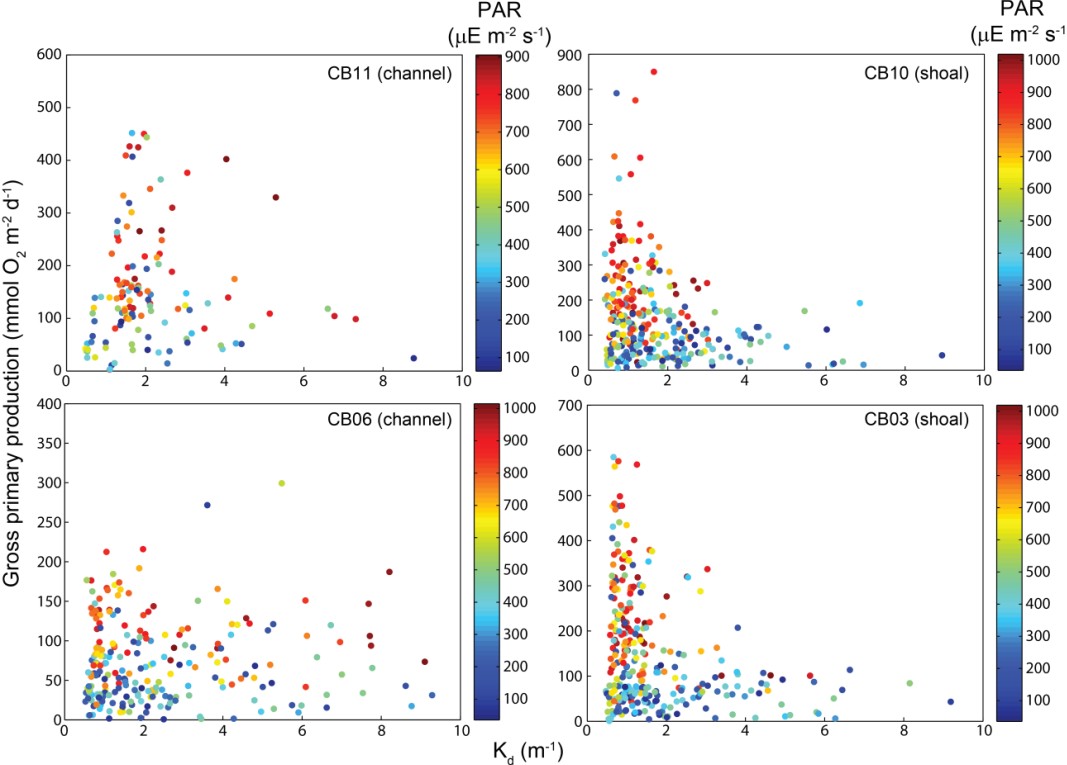

Figure 12. Relationship between light attenuation $K_{dPAR}$ and gross primary production, with coloration indicating

surface photosynthetically active radiation (PAR, $\mu M\ m^{-2}\ s^{-1}$) measured at the weather station. Significant reduction

in gross primary production at the highest $K_{dPAR}$ is observed at vegetated shoal sites.

