# Peer review of "Spatiotemporal variability of light attenuation and net"

_Biogeosciences, 2018_

## Referee Comment (RC1) · Anonymous Referee #1 · 6 Aug 2018

The manuscript "Spatiotemporal variability of light attenuation and net ecosystem metabolism in a back-barrier estuary" presents the results of a comprehensive water quality sampling program situated in Chincoteague Bay, Maryland/Virginia. The manuscript is well-written and all results are presented clearly.

There are three main concerns I have with the manuscript as it currently stands:

1. There appears to be no clear conclusion apart from the point that measuring quantities with high spatiotemporal resolution is useful – so the manuscript in its current form lacks novelty.

[Figure]

For example, consider the last sentence of the Introduction, "Our conclusions highlight the importance of quantifying spatiotemporal variability in these processes, which indicate feedbacks between physical and ecological processes in marine environments that should be considered when evaluating future ecosystem response." However, there is no explicit consideration of feedbacks in the manuscript apart from a brief mention in the Discussion.

Alternatively, consider the last sentence of the Abstract, "This study demonstrates how extensive continuous physical and biological measurements can help determine metabolic properties in a shallow estuary, including differences in metabolism and oxygen variability between SAV and phytoplankton-dominated habitats." The first half of this sentence is a self-evident point, but regarding the second half of this sentence, there is no specific quantitative analysis in the paper comparing sites that are SAV- and phytoplankton-dominated.

2. Time series data presented in the manuscript has already been published in a technical report available online – this would be fine if there was sufficient quantitative analysis of this data (see next point), but the figures showing these time series data also do not explicitly acknowledge that this data is already published elsewhere. Most of the data presented in Figures 2-5 and 7 is identical to data presented in Figures 46, 49, 50, 52-56 of the technical report Suttles et al. (2017) cited within the manuscript. Furthermore, there appear to be other water quality stations present at the study site (see Figures 2 and 3 of Suttles et al. 2017) that measured relevant quantities during the time periods of the study whose data was not considered in the manuscript – the reasons for this also need to be addressed.

3. There is very little quantitative analysis of the results, and conclusions appear to be drawn from the presented figures without sufficient justification. Consider the first paragraph of the Results. The first sentence states that "Turbidity ranged from near zero to a maximum of over 400 NTU at site CB06 during a winter storm

that induced waves exceeding 0.7 m (Figs. 2-5)." However, there is no indication in the manuscript (or figures) of when this winter storm took place, and no data presented for wave heights. The second sentence states that "sites CB03, CB10, and CB11 had similar statistical distributions of turbidity", but there is no statistical analysis of turbidity present in the manuscript, only time series data.

If this manuscript were rewritten for future publication, one possible focus could be on the spectral signals shown in Figure 6 to potentially give advice to the broader scientific community regarding the temporal scales for which water quality quantities need to be measured in order to sufficiently capture their "true" values, e.g. for comparison between sites and/or time periods. Overall, the manuscript needs to go beyond the presented time series and undertake further statistical (or other relevant) analyses of these time series to reveal differences between sites. With such analysis, it may be possible from the excellent data, obtained from this monitoring program, to yield conclusions that are novel and broadly applicable to the scientific community.

---

## Short Comment (SC1) · 22 Oct 2018

The diel method typically assumes that oxygen is well mixed throughout the boundary layer (hbl). Only when this is true can the time rate of change of oxygen be "corrected" for the surface flux (Fsurf). Furthermore, the surface flux correction requires that there is no flux through the bottom of the mixed layer (or seabed in shallow water) so that the flux divergence can estimated simply as Fsurf/hbl. In shallow water, like the environments studied here, there could benthic fluxes that would invalidate this estimation. When the diel method is applied and these assumptions are not valid, diel variations in the flux divergence term are not accurately accounted for typically resulting in an

over-estimate of community respiration. This often can result in an apparent first order balance between GPP and CR (like in figures 9 and 11), when in reality the flux divergence is much more important than assumed. In the absence of advection, GPP+CR must be balanced by the time rate of change and the flux divergence. If the flux divergence term is poorly estimated by the bulk estimate, these errors will be included in the estimate of CR resulting in a nearly 1:1 relationship between GPP and CR (like in figure 11). I think that the estimated flux divergence term should be shown so that the reader knows how big this term is compared to the estimated NEM. I would not be surprised if the errors associated with the estimated flux divergence are larger than the estimates of NEM. In my experience, diel methods provide useful estimates of GPP but are not accurate enough to resolve NEM. I think some comments should be added regarding whether or not vertical oxygen gradient develop and the potential role of benthic fluxes. In addition to these errors, there is considerable uncertainty in the piston velocity in these systems. I assume a wind-speed dependent formulation was used, but this should be discussed more explicitly, including a discussion its applicability to a sheltered estuarine environment.

---

## Referee Comment (RC2) · Anonymous Referee #2 · 6 Nov 2018

This paper investigated spatio-temporal variability of light attenuation, of Pgros and R in Chincoteague Bay, Maryland/Virginia, USA. The authors have valuable long-term data and the paper is overall well-written, however I have some concerns on this version of the manuscript.

Firstly, the result section could be substantially improved. As it is, there is limited quantitative data. For instance, ranges and means of Pgross, Pnet and R should be clearly reported.

To calculate Pnet, Pgross and R the authors need flow speed and direction, oxygen, PAR, and wind data. I assume wind data were used in the calculations, however the

authors should explicitly report the formula and variables used in their calculations.

Also, the authors have a valuable long-term data-set. My main concerns is that the instruments recorded data every 15 min, which is not an ideal resolution, it would be good to comment on this.

The conclusions and overall significance of the study could also be improved substantially. There is not a clear and compelling "take home message" at the moment.

Importantly, a great limitation of many studies on community metabolism is the lack of data on community structure. The authors could extrapolate data on benthic communities in their study sites and discuss the role of community structure in greater detail, this would strengthen the paper.

---

## Author Comment (AC1) · 13 Dec 2018

**bg-2018-335: "Spatiotemporal variability of light attenuation and net ecosystem metabolism in a back-barrier estuary"**

**by Neil K. Ganju, Jeremy M. Testa, Steven E. Suttles, and Alfredo L. Aretxabaleta**

**Response to Reviewers, comments in plain text, response in bold**

**Anonymous Referee #1**

The manuscript "Spatiotemporal variability of light attenuation and net ecosystem metabolism in a back-barrier estuary" presents the results of a comprehensive water quality sampling program situated in Chincoteague Bay, Maryland/Virginia. The manuscript is well-written and all results are presented clearly. There are three main concerns I have with the manuscript as it currently stands:

1. There appears to be no clear conclusion apart from the point that measuring quantities with high spatiotemporal resolution is useful – so the manuscript in its current form lacks novelty. For example, consider the last sentence of the Introduction, "Our conclusions highlight the importance of quantifying spatiotemporal variability in these processes, which indicate feedbacks between physical and ecological processes in marine environments that should be considered when evaluating future ecosystem response." However, there is no explicit consideration of feedbacks in the manuscript apart from a brief mention in the Discussion.

**In the abstract and elsewhere, we failed to include the feedback between SAV density and resuspension, which was quantified and discussed in Fig. 8 and Sec. 4.1. We intend to more fully integrate this result into the abstract and conclusion, as well as the new analyses discussed below.**

**We have expanded the analysis to quantitatively address the influence of temporal resolution on discerning spatial gradients (see below), and conducted a wavelet analysis to quantify difference between SAV and non-SAV sites (also see below). We believe these two new analyses make appropriate use of the data to inform future studies.**

Alternatively, consider the last sentence of the Abstract, "This study demonstrates how extensive continuous physical and biological measurements can help determine metabolic properties in a shallow estuary, including differences in metabolism and oxygen variability between SAV and phytoplankton-dominated habitats." The first half of this sentence is a self-evident point, but regarding the second half of this sentence, there is no specific quantitative analysis in the paper comparing sites that are SAV- and phytoplankton-dominated.

**Regarding that sentence specifically, we can revise to:**

> **"This study quantifies differences in the timescales of co-variation of key water-column properties that represent controls on light availability and ecosystem metabolism in a back barrier estuary. The analysis reveals how light availability and ecosystem metabolism varies across habitats spanning nutrient-enrichment gradients and different dominant primary producers. Through these analyses, we document the dynamics of self-reinforcing growth feedbacks associated with marcrophyte-induced decreases in suspended particles and associated light attenuation."**

In addition, we will include more in-depth analysis of the spectral results, specifically noting the following:

1) With regards to dissolved oxygen, there is a stronger diurnal signal at shoal sites relative to channel sites, indicating higher local production/respiration
2) With regards to turbidity, the highest spectral density is in the low-frequency band at CB06, which demonstrates the spatial integration of resuspension process throughout the estuary at this main channel site.

3) With regards to chlorophyll, CB11, the eutrophic site, has highest low-frequency spectral density due to increased nitrogen inputs and eutrophication.

4) With regards to fDOM, the two northern sites (CB10, CB11) have highest low-frequency spectral density, indicating a spatial gradient in freshwater input from north to south.

We will also include a more involved wavelet analysis to quantitatively link the processes. For example, wavelet coherence between dissolved oxygen (DO) and turbidity indicates how resuspension processes and light attenuation influence production/respiration. We find that shoal sites have a stronger coherence between DO and turbidity over multi-day frequencies than channel sites, indicating a de-coupling of sediment transport and biogeochemical processes in the channels (Fig. AC1). We also find a strong coherence between chl-a and turbidity at all sites, suggesting that the chl-a signal during periods with increased wind-wave forcing may be mostly resuspension of benthic microalgae.

[Figure]

**Figure AC1. Wavelet coherence between dissolved oxygen (%) and turbidity at channel site CB06 (upper) and shoal site CB03 (lower). Increased coherence at CB03 for periods between ~128 and 2048 correspond to timescale of 1 to 21 days (respectively), and suggest a coupling between resuspension, light availability, and production/respiration that is not observed at channel sites.**

We have also resampled the four main water quality parameters (DO, turbidity, chlorophyll, and fDOM) at 1-h, 2-h, and daily intervals, to explore how temporal resolution affects means, minima, and maxima (Table AC1; Fig. AC2). We observe that mean values are relatively insensitive to sampling interval, however maximum values are significantly modulated for all parameters, while minimum values for DO are significantly modulated as well. With regards to dissolved oxygen specifically, we find that daily sampling dampens the spatial variability in maxima and minima between sites. This is an important finding, given the ubiquitous daily sampling programs in many estuaries which cover many sites. This result suggests that characterizing differences in water-column conditions across space requires sampling at timescales finer than 1 day, especially in highly metabolic environments.

[Figure]

Figure AC2. Mean (dots), maxima and minima (bars) for each parameter using different temporal sampling intervals (15 min, 60 min, 120 min, 1d). Spatial gradients in dissolved oxygen are most impacted by coarse temporal resolution, with differences in minima and maxima largely eliminated at resolution of 1 d.

**Table AC1. Mean, minima, and maxima for four water-quality parameters at four sites.**

| | | DO (%) | | | TURB (NTU) | | | Chl-a (ug/L) | | | fDOM (QSU) | | |
|---|---|---|---|---|---|---|---|---|---|---|---|---|---|
| | | mean | min | max | mean | min | max | mean | min | max | mean | min | max |
| CB03 | 15 min | 104.4 | 40.9 | 182.8 | 14.9 | 0.0 | 427.4 | 5.2 | 0.2 | 29.9 | 12.6 | 0.0 | 30.4 |
| | 60 min | 104.4 | 41.2 | 182.3 | 14.9 | 0.0 | 396.9 | 5.2 | 0.2 | 29.7 | 12.6 | 0.0 | 29.7 |
| | 120 min | 104.4 | 44.4 | 180.1 | 14.9 | 0.0 | 396.9 | 5.2 | 0.2 | 29.0 | 12.6 | 3.4 | 29.0 |
| | 1 day | 106.5 | 59.1 | 149.2 | 16.7 | 0.0 | 242.7 | 5.5 | 0.2 | 27.6 | 12.5 | 3.7 | 25.8 |
| CB06 | 15 min | 103.0 | 70.2 | 149.4 | 27.1 | 0.0 | 546.8 | 5.6 | 0.0 | 30.0 | 6.7 | 0.0 | 18.6 |
| | 60 min | 103.0 | 75.0 | 149.0 | 27.0 | 0.0 | 546.8 | 5.6 | 0.0 | 29.7 | 6.7 | 0.0 | 18.2 |
| | 120 min | 103.0 | 77.9 | 149.0 | 27.2 | 0.0 | 525.7 | 5.6 | 0.1 | 29.1 | 6.7 | 0.0 | 17.5 |
| | 1 day | 104.2 | 92.8 | 141.5 | 28.7 | 0.0 | 356.7 | 5.2 | 0.2 | 25.4 | 6.7 | 0.0 | 17.0 |
| CB10 | 15 min | 108.1 | 53.4 | 214.5 | 13.6 | 0.0 | 375.0 | 5.9 | 0.2 | 39.5 | 16.7 | 3.4 | 39.7 |
| | 60 min | 108.1 | 55.3 | 212.0 | 13.6 | 0.0 | 308.8 | 5.9 | 0.3 | 35.9 | 16.7 | 3.4 | 39.7 |
| | 120 min | 108.1 | 56.3 | 212.0 | 13.6 | 0.0 | 308.8 | 5.8 | 0.3 | 35.8 | 16.7 | 3.4 | 39.7 |
| | 1 day | 105.1 | 68.7 | 151.9 | 15.7 | 0.0 | 274.5 | 5.0 | 0.3 | 35.8 | 16.5 | 4.2 | 37.7 |
| CB11 | 15 min | 99.3 | 19.2 | 160.3 | 12.4 | 0.0 | 275.2 | 11.0 | 0.5 | 49.6 | 26.0 | 5.4 | 56.8 |
| | 60 min | 99.3 | 19.4 | 160.3 | 12.4 | 0.0 | 255.5 | 11.1 | 0.5 | 49.0 | 26.0 | 5.5 | 56.8 |
| | 120 min | 99.3 | 20.9 | 160.3 | 12.4 | 0.0 | 255.5 | 11.1 | 0.5 | 40.9 | 26.1 | 5.5 | 52.3 |
| | 1 day | 104.7 | 55.3 | 143.9 | 14.2 | 0.0 | 174.3 | 10.8 | 0.7 | 36.8 | 25.6 | 8.1 | 46.9 |

2. Time series data presented in the manuscript has already been published in a technical report available online – this would be fine if there was sufficient quantitative analysis of this data (see next point), but the figures showing these time series data also do not explicitly acknowledge that this data is already published elsewhere. Most of the data presented in Figures 2-5 and 7 is identical to data presented in Figures 46, 49, 50, 52-56 of the technical report Suttles et al. (2017) cited within the manuscript. Furthermore, there appear to be other water quality stations present at the study site (see Figures 2 and 3 of Suttles et al. 2017) that measured relevant quantities during the time periods of the study whose data was not considered in the manuscript – the reasons for this also need to be addressed.

**USGS policy requires release of data before submission of a peer-reviewed journal article. The report of Suttles et al. is a non-interpretive data release report that contains figures simply to connect the distributed data with time-series plots to ensure consistency. Every peer-reviewed journal article submitted by USGS authors has an associated data release. The figures are not identical to figures in the data release; they have been generated for this publication.**

**We only used the sites from our field campaign that allowed a comparison between SAV-dominated sites with light measurements (of which there were only two, CB03 and CB10), along with partner non-SAV sites that had continuous water quality measurements (of which there were only two, CB06 and CB11). The other sites only had velocity, waves,**

**and/or suspended-sediment measurements, which means they could not be used to estimate light attenuation.**

3. There is very little quantitative analysis of the results, and conclusions appear to be drawn from the presented figures without sufficient justification. Consider the first paragraph of the Results. The first sentence states that "Turbidity ranged from near zero to a maximum of over 400 NTU at site CB06 during a winter storm that induced waves exceeding 0.7 m (Figs. 2-5)." However, there is no indication in the manuscript (or figures) of when this winter storm took place, and no data presented for wave heights. The second sentence states that "sites CB03, CB10, and CB11 had similar statistical distributions of turbidity", but there is no statistical analysis of turbidity present in the manuscript, only time series data.

**We will clarify such details throughout, and will add wave data to the figures. We have performed more thorough statistical analyses for all parameters and sites, see above.**

If this manuscript were rewritten for future publication, one possible focus could be on the spectral signals shown in Figure 6 to potentially give advice to the broader scientific community regarding the temporal scales for which water quality quantities need to be measured in order to sufficiently capture their "true" values, e.g. for comparison between sites and/or time periods.

**We intend to add two main points to the discussion, presenting the wavelet analysis results and the influence of temporal sampling resolution on interpreting spatial differences, as indicated above.**

Overall, the manuscript needs to go beyond the presented time series and undertake further statistical (or other relevant) analyses of these time series to reveal differences between sites. With such analysis, it may be possible from the excellent data, obtained from this monitoring program, to yield conclusions that are novel and broadly applicable to the scientific community.

**We believe that the additional statistical analyses and wavelet coherence analyses have yielded new linkages that highlight the differences between the sites.**

Anonymous Referee #2

This paper investigated spatio-temporal variability of light attenuation, of Pgros and R in Chincoteague Bay, Maryland/Virginia, USA. The authors have valuable long-term data and the paper is overall well-written, however I have some concerns on this version of the manuscript. Firstly, the result section could be substantially improved. As it is, there is limited quantitative data. For instance, ranges and means of Pgross, Pnet and R should be clearly reported.

**We did include means and standard deviations of all derived metabolic rates in Table 2 for the August to September period. We will include a new Table, either in the main body or as supplemental material (Table AC2), that summarizes more details of these estimates over the entire year and we will include the key features of these values in the results.**

**Table AC2. Summary of metabolic rate estimates at the four study sites, including monthly means, standard deviation, minimum and maximum values. All rates in mmol $O_2$ $m^{-2}$ $d^{-1}$.**

| | | CB03 | | | CB06 | | | CB10 | | | CB11 | | |
| --- | --- | --- | --- | --- | --- | --- | --- | --- | --- | --- | --- | --- | --- |
| | | $P_g$ | $R_t$ | $P_n$ | $P_g$ | $R_t$ | $P_n$ | $P_g$ | $R_t$ | $P_n$ | $P_g$ | $R_t$ | $P_n$ |
| January | Mean (±SD) | 44.76 (±35) | 41.69 (±35.8) | 3.07 (±24.81) | 11.95 (±52.79) | 4.62 (±65.26) | 7.33 (±28.07) | 37.27 (±73.44) | 31.55 (±82.98) | 5.72 (±27.19) | 52.84 (±9.6) | 36.74 (±8.61) | 16.10 (±7.05) |
| | Minimum | 4.62 | 2.85 | -46.02 | 1.33 | 1.30 | -51.79 | 13.62 | 10.10 | -51.74 | 37.35 | 25.02 | 6.57 |
| | Maximum | 113.65 | 104.39 | 53.92 | 84.29 | 109.13 | 66.12 | 166.57 | 196.32 | 58.50 | 64.22 | 43.15 | 25.95 |
| February | Mean (±SD) | 21.73 (±17.9) | 13.00 (±19.76) | 8.73 (±8.85) | 17.72 (±20.27) | 9.70 (±22.4) | 8.02 (±12.17) | 46.86 (±29.53) | 41.28 (±34.07) | 5.58 (±9.51) | 63.13 (±38.38) | 55.32 (±42.32) | 7.81 (±10.42) |
| | Minimum | 0.74 | 1.15 | -6.56 | 0.66 | 6.64 | -14.04 | 10.88 | 5.29 | -12.57 | 11.08 | 10.39 | -11.98 |
| | Maximum | 54.19 | 48.82 | 30.95 | 56.86 | 47.67 | 50.23 | 122.55 | 127.75 | 40.23 | 140.83 | 152.81 | 43.97 |
| March | Mean (±SD) | 61.25 (±61.4) | 61.61 (±55.28) | -0.36 (±21.39) | 52.45 (±34.92) | 54.98 (±44.06) | -2.54 (±20.4) | 36.12 (±75.76) | 34.98 (±70.46) | 1.14 (±14.17) | 133.50 (±27.64) | 139.12 (±25.72) | -5.62 (±15.55) |
| | Minimum | 28.28 | 2.55 | -66.70 | 15.26 | 6.67 | -71.49 | 17.93 | 12.35 | -41.94 | 87.15 | 91.41 | -29.87 |
| | Maximum | 162.50 | 145.89 | 31.04 | 121.08 | 185.61 | 23.68 | 121.63 | 108.83 | 44.34 | 198.99 | 196.66 | 26.00 |
| April | Mean (±SD) | 125.06 (±72.4) | 114.54 (±72.17) | 10.53 (±19.04) | 106.12 (±72.98) | 113.17 (±73.74) | -7.04 (±27.87) | 129.30 (±51.3) | 126.15 (±47.51) | 3.15 (±34.1) | 122.29 (±38.18) | 116.49 (±42.53) | 5.80 (±17.13) |
| | Minimum | 19.49 | 3.26 | -16.43 | 38.91 | 12.37 | -96.89 | 59.40 | 50.83 | -137.79 | 45.66 | 29.54 | -25.89 |
| | Maximum | 284.89 | 263.19 | 86.13 | 299.18 | 327.50 | 35.00 | 238.54 | 238.84 | 58.45 | 196.30 | 197.49 | 33.48 |
| May | Mean (±SD) | 232.09 (±84.76) | 221.45 (±87.63) | 10.64 (±19.57) | 113.71 (±68.60) | 114.93 (±62.85) | -1.21 (±12.9) | 233.96 (±106.35) | 226.60 (±101.84) | 7.36 (±12.33) | N/A | N/A | N/A |
| | Minimum | 100.94 | 67.31 | -33.61 | 71.46 | 51.59 | -32.32 | 22.94 | 35.40 | -16.00 | N/A | N/A | N/A |
| | Maximum | 359.46 | 355.89 | 57.05 | 215.87 | 219.11 | 22.63 | 381.95 | 369.99 | 30.21 | N/A | N/A | N/A |
| June | Mean (±SD) | 237.11 (±85.29) | 232.17 (±84.72) | 4.94 (±14.39) | 109.72 (±15.14) | 104.53 (±34.87) | 5.19 (±23.74) | 232.80 (±87.72) | 231.43 (±89.33) | 1.38 (±19.19) | N/A | N/A | N/A |
| | Minimum | 59.70 | 82.49 | -22.79 | 91.22 | 72.36 | -52.86 | 93.25 | 122.96 | -32.17 | N/A | N/A | N/A |
| | Maximum | 401.62 | 393.89 | 55.29 | 136.62 | 189.48 | 24.41 | 416.03 | 400.18 | 39.66 | N/A | N/A | N/A |
| July | Mean (±SD) | 182.72 (±82.9) | 183.56 (±75.85) | -0.83 (±12.4) | N/A | N/A | N/A | 179.69 (±138.97) | 173.03 (±124.99) | 6.66 (±30.83) | N/A | N/A | N/A |
| | Minimum | 56.72 | 73.34 | -18.12 | N/A | N/A | N/A | 75.63 | 0.35 | -43.01 | N/A | N/A | N/A |
| | Maximum | 302.17 | 299.71 | 26.33 | N/A | N/A | N/A | 381.61 | 373.88 | 74.33 | N/A | N/A | N/A |
| August | Mean (±SD) | 344.85 (±154.49) | 355.40 (±157.25) | -10.55 (±23.19) | 100.66 (±56.15) | 105.91 (±38.33) | -5.24 (±41.22) | 334.79 (±256.96) | 323.77 (±246.55) | 11.02 (±45.9) | 289.36 (±124.99) | 317.37 (±94.08) | -28.01 (±93.62) |
| | Minimum | 129.45 | 139.56 | -51.69 | 59.12 | 33.63 | -145.80 | 81.41 | 24.84 | -115.35 | 188.52 | 199.70 | -372.23 |
| | Maximum | 575.94 | 612.93 | 37.94 | 162.47 | 173.16 | 34.20 | 850.14 | 812.87 | 111.49 | 449.78 | 492.57 | 21.63 |
| September | Mean (±SD) | 292.56 (±165.9) | 276.69 (±187.30) | 15.87 (±39.71) | 89.22 (±52.93) | 85.89 (±61.99) | 3.32 (±15.19) | 253.74 (±157.03) | 236.73 (±163.69) | 17.01 (±47.56) | 213.65 (±132.99) | 217.51 (±139.56) | -3.87 (±20.89) |

| | | | | | | | | | | | | | |
|---|---|---|---|---|---|---|---|---|---|---|---|---|---|
| | Minimum | 44.69 | 131.45 | -38.05 | 5.71 | 28.71 | -32.78 | 7.76 | 74.83 | -76.46 | 2.67 | 28.04 | -49.79 |
| | Maximum | 584.92 | 564.66 | 145.06 | 165.74 | 173.93 | 47.45 | 788.71 | 795.48 | 117.17 | 451.94 | 479.71 | 35.39 |
| October | Mean (±SD) | 133.08 (±84.29) | 129.62 (±69.26) | 3.46 (±42.05) | 34.63 (±49.0) | 23.80 (±35.88) | 10.83 (±32.75) | 82.91 (±61.18) | 69.74 (±62.71) | 13.17 (±20.6) | 88.68 (±45.05) | 81.93 (±35.62) | 6.75 (±23.29) |
| | Minimum | 53.3 | 5.24 | -154.65 | 9.59 | 9.41 | -17.02 | 18.22 | 7.93 | -29.23 | 53.90 | 31.59 | -54.24 |
| | Maximum | 323.11 | 268.09 | 76.23 | 176.73 | 74.83 | 158.44 | 191.25 | 178.44 | 42.91 | 209.12 | 171.12 | 40.71 |
| November | Mean (±SD) | 67.45 (±19.42) | 57.79 (±15.62) | 9.66 (±35.8) | 25.14 (±16.76) | 17.50 (±16.32) | 7.64 (±9.24) | 81.76 (±40.59) | 71.52 (±47.23) | 10.23 (±12.92) | 47.54 (±42.61) | 36.88 (±50.24) | 10.65 (±17.83) |
| | Minimum | 34.28 | 29.37 | -2.89 | 1.69 | 1.23 | -5.10 | 21.77 | 13.85 | -7.94 | 0.86 | 1.59 | -20.65 |
| | Maximum | 120.15 | 93.52 | 34.59 | 60.97 | 54.32 | 27.83 | 155.66 | 159.77 | 41.03 | 97.75 | 99.72 | 43.32 |
| December | Mean (±SD) | 54.19 (±32.97) | 45.32 (±37.68) | 8.87 (±19.14) | 30.01 (±32.34) | 27.84 (±33.34) | 2.17 (±6.45) | 58.31 (±44.15) | 50.12 (±43.67) | 8.19 (±16.16) | N/A | N/A | N/A |
| | Minimum | 15.58 | 12.93 | -59.61 | 2.14 | 6.02 | -10.98 | 6.08 | 3.75 | -21.36 | N/A | N/A | N/A |
| | Maximum | 123.94 | 128.21 | 52.98 | 79.63 | 77.46 | 15.25 | 131.48 | 136.35 | 55.04 | N/A | N/A | N/A |

To calculate Pnet, Pgross and R the authors need flow speed and direction, oxygen, PAR, and wind data. I assume wind data were used in the calculations, however the authors should explicitly report the formula and variables used in their calculations.

**We agree that this is an important detail of the calculation and will add this information in the methods section of the revised manuscript.**

> *"The changes in dissolved oxygen concentrations used to compute metabolic rates were corrected for air-water gas exchange using the equation $D = k_a(C_s-C)$, where D is the rate of air-water oxygen exchange (mg $O_2$ $L^{-1}$ $h^{-1}$), $K_a$ volumetric aeration coefficient ($h^{-1}$), and $C_s$ and C are the oxygen saturation concentration and observed oxygen concentration (mg $O_2$ $L^{-1}$), respectively. $K_a$ was computed as a function of wind speed measured at a weather station installed at a dock at Public Landing, Maryland (western shore of Chincoteague Bay near station 10) during the course of the sensor deployment. Details of the air-water gas calculation are incorporated into the R package WtRegDO (Beck et al. 2015) and described in detail in Thebault et al. (2008), and we utilized atmospheric pressure and air temperature data for these calculations from a nearby buoy (OCIM2 - 8570283 at the Ocean City Inlet, Maryland)."*

Also, the authors have a valuable long-term data-set. My main concerns is that the instruments recorded data every 15 min, which is not an ideal resolution, it would be good to comment on this.

**It is not clear why 15 min resolution is not ideal. To our knowledge, there are few, if any studies that examine time-series at these short- time scales for waves, water-column physical and biogeochemical variables, and light availability. While a subset of these variables may have been examined in previous estuarine studies, The comprehensive nature of this continuously-measured variable suite is unique. At this timescale, all tidal**

**variations are resolved; wave measurements are collected with burst samples that resolve waves with periods between approximately 1 and 15 seconds. And our newly included statistical analysis (above) shows that mean/minimum/maximum values with 1 and 2 h sampling interval are similar.**

The conclusions and overall significance of the study could also be improved substantially.

**We believe the conclusions and significance have been enhanced with our new proposed analyses (see above).**

There is not a clear and compelling "take home message" at the moment. Importantly, a great limitation of many studies on community metabolism is the lack of data on community structure. The authors could extrapolate data on benthic communities in their study sites and discuss the role of community structure in greater detail, this would strengthen the paper.

**We unfortunately do not have data of this type to combine with our measurements. The SAV beds we studied were uniformly *composed of Zostera marina,* but the macroalgae and epiphytes that also occupied the habitat were not quantified and identified, as this was outside of the scope of our study. We agree, however, that we should make clearer statements regarding what is compelling about our analysis. In the revised manuscript, we now state how our analysis clearly quantifies the (1) the timescales in which dissolved oxygen dynamics correlate to factors driving light availability, (2) how the strength of relationships between physical and biological properties varies between vegetated and non-vegetated habitats, (3) the identification of the primary drivers of light attenuation in the different habitats, and (4) how ecosystem metabolism varies across habitats spanning nutrient-enrichment gradients and different dominant primary producers.**

**Our two proposed analyses, using wavelet coherence and resampling of means, maxima, and minima, explore the differences between channel and shoal sites, and the influence of temporal resolution on interpretation of spatial gradients.**

---

## Author Comment (AC2) · 13 Dec 2018

Please see uploaded pdf, "bg-2018-335-supplement.pdf"

Please also note the supplement to this comment:
https://www.biogeosciences-discuss.net/bg-2018-335/bg-2018-335-AC2-supplement.pdf

———————————————

---

## Author Comment (AC3) · 13 Dec 2018

**bg-2018-335: "Spatiotemporal variability of light attenuation and net ecosystem metabolism in a back-barrier estuary"**

**by Neil K. Ganju, Jeremy M. Testa, Steven E. Suttles, and Alfredo L. Aretxabaleta**

**Response to M. Scully, comments in plain text, response in bold**

Comment: M. Scully

The diel method typically assumes that oxygen is well mixed throughout the boundary layer (hbl). Only when this is true can the time rate of change of oxygen be "corrected" for the surface flux (Fsurf). Furthermore, the surface flux correction requires that there is no flux through the bottom of the mixed layer (or seabed in shallow water) so that the flux divergence can estimated simply as Fsurf/hbl. In shallow water, like the environments studied here, there could benthic fluxes that would invalidate this estimation. When the diel method is applied and these assumptions are not valid, diel variations in the flux divergence term are not accurately accounted for typically resulting in an over-estimate of community respiration. This often can result in an apparent first order balance between GPP and CR (like in figures 9 and 11), when in reality the flux divergence is much more important than assumed. In the absence of advection, GPP+CR must be balanced by the time rate of change and the flux divergence. If the flux divergence term is poorly estimated by the bulk estimate, these errors will be included in the estimate of CR resulting in a nearly 1:1 relationship between GPP and CR (like in figure 11). I think that the estimated flux divergence term should be shown so that the reader knows how big this term is compared to the estimated NEM. I would not be surprised if the errors associated with the estimated flux divergence are larger than the estimates of NEM. In my experience, diel methods provide useful estimates of GPP but are not accurate enough to resolve NEM. I think some comments should be added regarding whether or not vertical oxygen gradient develop and the potential role of benthic fluxes. In addition to these errors, there is considerable uncertainty in the piston velocity in these systems. I assume a wind-speed dependent formulation was used, but this should be discussed more explicitly, including a discussion its applicability to a sheltered estuarine environment.

**This comment correctly points out that we did not specifically address vertical oxygen gradients at our study sites and that we did not attempt to quantify flux divergence. There are several features of our approach that could result in contrasts with questions raised here related to flux divergence. First, our study sites are less than or equal to 3 meters deep, and although we do not have vertical profile data for dissolved oxygen at the ~1 m SAV-dominated sites, we think it is a reasonable assumption to conclude that these waters are vertically well mixed with respect to oxygen. Secondly, we deployed our sensors just above the sediment surface, so they are less vulnerable to issues of flux divergence associated with a sensor being placed in the surface water where elevated light availability will drive higher diel variations in oxygen relative to underlying water. Third, two of our sites are SAV dominated and the non-SAV sites have mean kd values that would allow > 1% surface light to reach the bottom, so benthic primary production is either dominant or likely at our sites. For this reason, we deployed our sensors near the bottom to capture both benthic and water column primary production. This feature either**

avoids or confounds issues of oxygen flux through the bottom of the mixed layer, because these benthic oxygen fluxes are the primary metabolic signal we aimed to measure. For our two deeper sites, which were 2-3 m, we were able to access vertical profile data for dissolved oxygen, temperature, and salinity that was collected monthly as part of monitoring performed by the Maryland Department of Natural Resources from 1999-2014. These data do indicate vertical dissolved oxygen differences of >1 mg/L on occasion over the 15 year record, especially during the productive summer months (Fig. SC1). The long-term mean vertical oxygen difference at the two sites was less than 0.5 mg/L during September to May, but between 0.5 and 0.9 mg/L during June-August. These observations suggest that our computations at the two deeper sites could fail the assumption of complete vertical mixing on some occasions, and we will address this potential limitation in the revised manuscript. This limitation appears to arise substantially in only July and August.

We agree that NEM is difficult to quantify, and aside from issues of flux divergence, NEP is computed as a small difference between the estimates of GPP and R, so its value relative to potential error is small. For this reason, we minimized discussion of NEM to a limited part of the manuscript and instead focused on GPP and R. Finally, we agree that details of the wind-speed formulation require further discussion and detail, which we will provide in the revised manuscript in response to another reviewer comment.

[Figure]

Figure SC1. Computed differences between surface and bottom dissolved oxygen (red circles), water temperature (blue circles), and salinity (green circles) at two stations in Chincoteague Bay from 1999-2014. Vertical differences are only reported when enough data were available to compute differences over more than 0.9 meters. Data collected by the Maryland Department of Natural Resources. Newport Bay is near CB11 (XCM4878) and the site called Channel near Public Landing, MD is near CB06 (XBM8149), please see http://eyesonthebay.dnr.maryland.gov/